# Plan2Cleanse: Test-Time Backdoor Defense via Monte-Carlo Planning in Deep Reinforcement Learning

**Sze-Ann Chen**♣, **Zhi-Yi Chin**♢, **Kui-Yuan Chen**♣, **Chi-Yu Li**♣, **Ping-Chun Hsieh**♣

♣**National Yang Ming Chiao Tung University, Hsinchu, Taiwan**
♢**CISPA Helmholtz Center for Information Security**
`pinghsieh@nycu.edu.tw`

Reviewed on OpenReview: `https://openreview.net/forum?id=ZKhKxqwuPu`

## Abstract

Ensuring the security of reinforcement learning (RL) models is critical, particularly when they are trained by third parties and deployed in real-world systems. Attackers can implant backdoors into these models, causing them to behave normally under typical conditions, but execute malicious behaviors when specific triggers are activated. In this work, we propose Plan2Cleanse, a test-time detection and mitigation framework that adapts Monte Carlo Tree Search to efficiently identify and neutralize RL backdoor attacks without requiring model retraining. Our approach recasts backdoor detection as a planning problem, enabling systematic exploration of temporally extended trigger sequences while maintaining black-box access to the target policy. By leveraging the detection results, Plan2Cleanse can further achieve efficient mitigation through tree-search preventive replanning. We evaluated our method in competitive MuJoCo environments, simulated O-RAN wireless networks, and Atari games. Plan2Cleanse achieves substantial improvements, increasing trigger detection success rates by more than 61.4 percentage points in stealthy O-RAN scenarios and improving win rates from 35% to 53% in competitive Humanoid environments. These results demonstrate the effectiveness of our test-time defense approach and highlight the importance of proactive defenses against backdoor threats in RL deployments. Our implementation is publicly available at `https://github.com/rl-bandits-lab/RL-Backdoor`.

## 1 Introduction

Reinforcement learning (RL) has achieved widespread adoption in domains such as games (Mnih et al., 2013; Silver et al., 2017), robotics (Kalashnikov et al., 2018; Zhu et al., 2020), and communication networks (Yu et al., 2019; Luong et al., 2019). However, training RL agents typically requires extensive interaction data, reward engineering, and hyperparameter tuning before deployment (Henderson et al., 2018; Adkins et al., 2024). To reduce development costs, practitioners increasingly adopt pre-trained RL models (Kumar et al., 2023; Yang et al., 2024; Reed et al., 2022; Zitkovich et al., 2023; Black et al., 2024; Sikchi et al., 2025; Wang et al., 2025) sourced from third-party vendors or public repositories. While this model-sharing paradigm accelerates integration, it also introduces security risks: malicious behaviors can be embedded into pre-trained models during training, posing severe threats in safety-critical systems where RL policies directly influence high-stakes decisions.

A notable security concern is *backdoor attacks*, where an adversary injects hidden triggers during training, enabling malicious behaviors to be activated at inference time. While backdoors in supervised learning commonly rely on static input perturbations (Gu et al., 2019; Liu et al., 2018b), RL introduces unique vulnerabilities due to its sequential and interactive nature. Actions influence future observations and rewards, allowing adversaries to design *temporally extended triggers* that activate only after specific state or action

sequences are executed (Wang et al., 2021b), often over tens or hundreds of timesteps. This temporal complexity greatly increases stealthiness and complicates both the detection and mitigation of RL backdoors.

The challenge is further exacerbated in black-box settings, where third-party pre-trained RL models provide no access to internal parameters or training data. As a result, defenders cannot perform white-box security analysis, such as inspecting model weights, gradients, or neuron activations to verify whether malicious behaviors are embedded. Such settings commonly arise when RL agents are deployed as large pre-trained foundation models, where parameter inspection or retraining is computationally infeasible, or when access is restricted to API-style query interfaces that expose only input-output behavior. This limited accessibility highlights the need for defense methods that operate without model retraining or weight modification. Notably, in safety-critical domains, it is standard practice in robustness and safety evaluation to stress-test learned policies in either simulators or controlled environments before real-world release, and the growing adoption of digital twin technologies, such as in robotics (Mittal et al., 2025), industrial control (Xu et al., 2023), and communication systems (Hoydis et al., 2022), also makes simulator access increasingly available as part of existing verification infrastructure.

Existing defense for RL backdoor attacks can be categorized into *fine-tuning* and *test-time* methods[1]. Fine-tuning methods (Wang et al., 2019; Chen et al., 2023; Guo et al., 2023; Yuan et al., 2024) require access to model weights or clean datasets, which can be unavailable for third-party models in practice. Moreover, these fine-tuning methods can be computationally demanding due to iterative policy updates over many episodes, especially when the pre-trained models are large. In contrast, test-time approaches can offer significant advantages: they require no model retraining or fine-tuning, can be applied to any pre-trained model regardless of its training process, and provide immediate deployment flexibility without modifying the parameters of the original policy model. However, existing test-time methods (Gao et al., 2019; Bharti et al., 2022) focus primarily on addressing one-step perturbation-based attacks and are not directly applicable to defense against the stealthy temporally extended backdoor triggers. As a result, there exists one critical and yet underexplored research question: **How to design a test-time backdoor defense method against temporally extended triggers in RL?**

To address this, we propose Plan2Cleanse, a planning-based defense framework designed for pre-deployment backdoor detection for safety evaluation and test-time backdoor mitigation as a safeguard mechanism. Plan2Cleanse fundamentally advances test-time detection and mitigation by eliminating neural policy fine-tuning altogether, instead employing Monte Carlo Tree Search (MCTS) with a Voronoi-based exploration strategy. Conceptually, we reinterpret backdoor detection as a tree search problem, framing the search for trigger action sequences as an optimization process over the action space. Regarding detection, this approach enables systematic traversal of the action space without gradient-based learning limitations, providing more comprehensive coverage of potential trigger sequences while maintaining the flexibility benefits of test-time approaches. Regarding mitigation, this approach can be augmented by short-horizon replanning to neutralize the attacks, without any fine-tuning. Since only short-horizon rollouts are used for relative trajectory comparison and local correction, the method can tolerate moderate simulator inaccuracies, making it compatible with approximate simulation or digital twin environments.

In summary, we make the following contributions:

- We formulate backdoor detection in RL as a trajectory-level planning problem and develop an MCTS-based framework that directly searches for adversarial trigger action sequences without relying on a separate probing policy, achieving over 99% detection success in Humanoid and substantially outperforming prior methods.
- We introduce a fully test-time mitigation mechanism that performs online short-horizon replanning to neutralize detected attacks while preserving benign performance, without modifying model parameters or requiring model retraining.
- We demonstrate the generality of Plan2Cleanse across diverse RL domains, including MuJoCo, O-RAN, and Atari, showing that a single planning-based test-time framework can defend against both temporally extended and observation-level backdoor triggers.

---

[1]Throughout the paper, we use the jargon "test-time method" to refer to a defense method that does not require any model retraining or fine-tuning, as usually adopted by the literature of pre-trained foundation models.

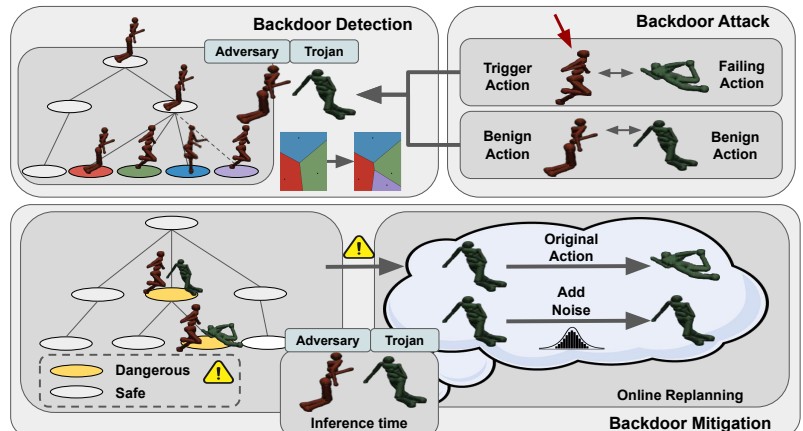

Figure 1: An overview of our Plan2Cleanse framework, which addresses backdoor vulnerabilities in RL through three components. (1) **Backdoor Attack:** During training, the adversary poisons the agent so that specific trigger actions cause it to switch from benign to failing behavior. (2) **Backdoor Detection:** MCTS explores the space of adversarial action sequences using Voronoi-guided sampling to efficiently identify candidate triggers, which are then validated through replay to confirm they reliably activate the backdoor. (3) **Backdoor Mitigation:** At test time, incoming adversary actions are checked against known trigger regions; when a match is detected, online replanning generates a safe alternative action without modifying the model parameters, preserving normal performance under benign conditions.

## 2 Related Works

### 2.1 Backdoor Attacks in RL

Backdoor attacks in RL can be divided into the following two major categories:

**Adversarial-Agent Attacks.** Given the sequential and interactive nature of RL, Wang et al. (2021b) introduces BackdooRL, which performs backdoor attacks in two-agent competitive settings, where a specific opponent's multi-step action sequence is utilized as a backdoor trigger and embedded into the policy during training such that the attacks can be triggered by an adversarial opponent at test time. Such an *interaction-level* attack is stealthy as it requires the defenders to recognize subtle multi-step patterns. Recent work has further diversified trigger designs, including temporal observation sequences (Yu et al., 2022) and sparse adaptive state poisoning approaches (Cui et al., 2024). More recently, (Liu et al., 2025) proposes supply-chain backdoor attacks, where an adversary with limited access implants backdoors through legitimate interactions with supplied agents or environment components.

**Perturbation-Based Attacks.** In parallel to interaction-level backdoors, several works adapt perturbation-based backdoor techniques from supervised learning (Gu et al., 2019; Liu et al., 2018b) to RL and introduce observation-level patch triggers for image-based control tasks (Kiourti et al., 2020; Wang et al., 2021c; Chen et al., 2022; Cui et al., 2024). These studies show that small observation-level patch triggers, injected during training, can later be activated at deployment to degrade the agent's performance. More recently, Rathbun et al. (2024) proposed a training-time poisoning framework that plants stealthy observation-level triggers across different RL algorithms and environments. In addition to the training-time attacks, Vyas et al. (2025) introduced two post-training attacks, namely (i) TrojanentRL, which embeds a backdoor via manipulation of the rollout buffer, and (ii) InfrectroRL, which injects backdoor behavior through direct, data-free modification of pretrained parameters.

### 2.2 Backdoor Detection and Mitigation in RL

**Defense Against Adversarial-Agent Attacks.** Adversarial-agent backdoors are difficult to detect because their activations are sparse, span multiple steps, and depend on specific interaction patterns rather

than a single observation (Wang et al., 2021b). Hence, supervised-learning style defenses that rely on observation-level anomaly detection or static pattern filtering are often insufficient in RL. To tackle this setting, Guo et al. (2023) propose PolicyCleanse, which learns a probing policy to actively search for reward-degrading action sequences and then removes the discovered triggers by fine-tuning the victim policy, *e.g.*, with PPO (Schulman et al., 2017; Huang et al., 2024). By contrast, our method is a test-time method that does not require any fine-tuning and hence obviates the need for access to the model weights.

**Defense Against Perturbation-Based Attacks.** To defend against perturbation-based backdoors, RL can leverage defenses originally developed for supervised learning, such as spectral signatures (Tran et al., 2018), activation-pattern inspection (Chen et al., 2019a), and pruning-based methods (Liu et al., 2018a), though these typically assume white-box access or clean data. To relax these assumptions, subsequent works propose black-box trigger recovery or test-time filtering, including Neural Cleanse (Wang et al., 2019), DeepInspect (Chen et al., 2019b), and STRIP (Gao et al., 2019). Building on RL-specific backdoor studies such as TrojDRL (Kiourti et al., 2020), subsequent defenses target observation-level triggers in image-based control: Provable Defense (PD) projects states into a safe subspace (Bharti et al., 2022), SHINE reconstructs patch triggers and retrains the policy to suppress them (Yuan et al., 2024), and BIRD regularizes the activation space during fine-tuning (Chen et al., 2023). Acharya et al. (2023) formulated Trojan detection as a meta-classification problem and trained a meta-classifier to assign a Trojan probability score based on features extracted from observations (termed universal Trojan signatures, or UTS in short), assuming access to a labeled dataset of benign and Trojaned agents. REStore (Le Roux et al., 2024) studies black-box backdoor defense in supervised learning and proposes to leverages rare-event simulation via Monte Carlo sampling to recover triggers. Vyas et al. (2024) focused on in-distribution triggers, *i.e.*, triggers that lie within the anticipated observation distribution and are thus harder to detect, and proposed defense operates in the neural activation space (NAS), identifying anomalous neuron activation patterns induced by backdoor triggers. These works together outline the observation-level defense landscape that our test-time, planning-based approach complements.

**Positioning of Plan2Cleanse**. To better position Plan2Cleanse, we provide a structured conceptual comparison with the representative backdoor defense methods in Table 1. We compare methods along five dimensions: (1) whether access to the policy parameters is required, (2) whether it operates purely at test time or requires fine-tuning the Trojan policy, (3) whether clean data are required, and (4-5) the types of backdoor attacks addressed, including adversarial-agent attacks and perturbation-based attacks.

As shown in Table 1, existing methods typically require either policy access, fine-tuning, or clean data, and often target a specific attack setting. In contrast, Plan2Cleanse operates solely at test time, without requiring access to policy parameters or clean data, and supports perturbation-based attacks within a unified detection framework. This comparison clarifies the distinct design choices of Plan2Cleanse and highlights its complementary strengths relative to prior defenses.

Table 1: Conceptual comparison between Plan2Cleanse and representative backdoor defense methods. We compare methods based on policy access requirements, test-time operation, the need for clean data, and the types of backdoor attacks they address (adversarial-agent and perturbation-based). Plan2Cleanse operates purely at test time without requiring policy access or clean data, while supporting both adversarial-agent and perturbation-based attacks within a unified framework.

| Method | Access to Model Parameters | Category | Clean Data | Adversarial-agent Attacks | Perturbation-based Attacks |
|---|---|---|---|---|---|
| Neural Cleanse (Wang et al., 2019) | Required | Fine-Tuning | Required | | • |
| PD (Bharti et al., 2022) | Not Required | Test-Time | Required | | • |
| BIRD (Chen et al., 2023) | Required | Fine-Tuning | Required | • | • |
| PolicyCleanse (Guo et al., 2023) | Required | Fine-Tuning | Not Required | • | |
| UTS (Acharya et al., 2023) | Required | Test-Time | Required | | • |
| REStore (Le Roux et al., 2024) | Not Required | Test-Time | Not Required | | • |
| NAS (Vyas et al., 2024) | Required | Test-Time | Required | | • |
| SHINE (Yuan et al., 2024) | Required | Fine-Tuning | Not Required | • | • |
| Plan2Cleanse (Ours) | Not Required | Test-Time | Not Required | • | • |

# 3 Preliminaries and Problem Formulation

In this section, we formally present the problem setting of backdoor attacks with temporally extended triggers considered in our study and the corresponding RL formulation. We first describe the two-agent Markov Game (Littman, 1994) that models the underlying backdoor attack mechanism and our threat model. We then describe the objective functions of backdoor detection and mitigation.

## 3.1 Two-Agent Markov Games

We start by modeling the environment from the perspective of the Trojan agent. Specifically, we consider a discounted two-agent Markov game $\mathcal{M} = (\mathcal{S}, \mathcal{A}, \mathcal{T}, \mathcal{R}, \gamma)$ with state space $\mathcal{S}$, joint action space $\mathcal{A} = \mathcal{A}^{\text{Trojan}} \times \mathcal{A}^{\text{adv}}$, transition function $\mathcal{T} : \mathcal{S} \times \mathcal{A}^{\text{Trojan}} \times \mathcal{A}^{\text{adv}} \to \Delta(\mathcal{S})$[2], reward function $\mathcal{R} : \mathcal{S} \times \mathcal{A}^{\text{Trojan}} \times \mathcal{A}^{\text{adv}} \to \mathbb{R}$, and discount factor $\gamma \in (0, 1]$. $\mathcal{A}^{\text{Trojan}}$ and $\mathcal{A}^{\text{adv}}$ denote the action spaces of the Trojan agent and the adversary, respectively. The reward is scalar-valued, as we focus solely on the return of the Trojan agent. This formulation is general in that it can capture the most prevalent RL backdoor attacks, including those with one-step perturbation-based triggers in single-agent settings[3] (Kiourti et al., 2020; Wang et al., 2019; Bharti et al., 2022; Chen et al., 2023) and the adversarial-agent attacks with temporally extended triggers in the two-agent competitive setting (Wang et al., 2021b; Guo et al., 2023).

## 3.2 Backdoor Attacks With Temporally Extended Triggers

Temporally extended backdoor triggers get activated only when the Trojan agent encounters specific action sequences, making the attack stealthy under standard evaluation. A common instance arises in two-agent competitive RL, where triggers are defined behaviorally through interactive action sequences. For example, the Trojan policy is activated to induce failure only after the opponent performs a particular sequence, remaining dormant until the condition is met. Following BackdooRL (Wang et al., 2021b), we model such behavior-sequence triggers by training a single network to imitate a mixture of fast-failing and winning trajectories. The trigger steers the policy toward failing behavior while keeping nominal behavior intact otherwise. The state space $\mathcal{S}$ can be partitioned according to the trigger activation status into two disjoint subsets, $\mathcal{S}^{\text{normal}}$ and $\mathcal{S}^{\text{triggered}}$, representing states in which the trigger condition is inactive or active, respectively. We define two policies: a benign policy $\pi^{\text{win}} : \mathcal{S}^{\text{normal}} \to \mathcal{A}^{\text{Trojan}}$ and a fast-failing policy $\pi^{\text{fail}} : \mathcal{S}^{\text{triggered}} \to \mathcal{A}^{\text{Trojan}}$. The Trojan policy $\pi^{\text{Trojan}}$ is then defined as the following piecewise policy:

$$\pi^{\text{Trojan}}(s) = \begin{cases} \pi^{\text{win}}(s), & s \in \mathcal{S}^{\text{normal}}, \\ \pi^{\text{fail}}(s), & s \in \mathcal{S}^{\text{triggered}}. \end{cases} \tag{1}$$

Since the trigger activation status is encoded in the state partition, the Trojan policy $\pi^{\text{Trojan}}$ remains Markovian and does not depend on trajectory history explicitly.

Observation-based patch triggers can be viewed as a special case with trigger length 1, as exemplified by TrojDRL (Kiourti et al., 2020), where training-time poisoning or reward shaping associates the patch with a poisoned action while preserving benign performance when the patch is absent. Both cases map a trigger condition to a distinct action mode, while keeping behavior benign otherwise.

## 3.3 Threat Model

We consider a threat model in a *black-box* RL setting, where the defender has no access to the internal parameters or the training data of the Trojan agent. Our defense operates at *test time*, without model retraining or access to clean data. Backdoors may be activated through observation-level triggers or temporally extended triggers. We describe the threat model from the perspectives of the adversary and the defender.

---

[2]Throughout the paper, for a set $\mathcal{X}$, we use $\Delta(\mathcal{X})$ to denote the set of all probability distributions over $\mathcal{X}$.

[3]For one-step perturbation-based backdoor attacks in the single-agent setting, the adversary action space $\mathcal{A}^{\text{adv}}$ can be modeled as selecting the presence and spatial placement of a trigger patch.

**Adversary.** The adversary injects poisoned data to implant a backdoor into the target RL policy, either during training (Kiourti et al., 2020; Wang et al., 2021b) or after training (Vyas et al., 2025). By exposing the agent to specific trigger patterns, either single-step or multi-step, the policy learns to execute a malicious behavior that causes significant trajectory-level performance degradation once the trigger appears. The agent behaves normally under clean conditions but deviates when the trigger is activated at deployment.

**Defender.** The defender interacts with the trained agent in a black-box manner, without access to its training data or internal parameters. The defender can query the agent and observe its states, actions, and rewards in a *potentially poisoned* environment. The defender has access to a simulator[4] or environment model that allows evaluation of candidate input sequences, but does not control the real adversary during deployment. The goal is to detect trigger-induced abnormal performance-degrading behaviors and mitigate them without degrading the agent's clean performance.

Domain-specific instantiations of the above adversary and defender in O-RAN, competitive control, and Atari are provided in Appendix A.

### 3.4 Learning Objectives of Backdoor Detection and Mitigation

**Backdoor Detection.** The goal is to discover a trigger sequence $a_{1:T}^{\mathrm{adv}}$ of the adversary, which activates the backdoor behavior in the Trojan policy, in a sample-efficient manner, *i.e.*, by using as few sampled environment rollouts as possible. Rather than relying on predefined ground-truth triggers, we define the detection objective as identifying sequences that cause the Trojan policy to exhibit degraded behavior. Specifically, a sequence is regarded as a valid trigger if it leads to low returns for the Trojan agent. To this end, we define the adversarial reward at each timestep as $r_t^{(-)} := (-1) \cdot \mathcal{R}(s_t, a_t^{\mathrm{Trojan}}, a_t^{\mathrm{adv}})$, and formulate the detection objective as finding an adversary's action sequence that maximizes the cumulative adversarial reward:

$$\max_{a_{1:T}^{\mathrm{adv}}} \ \mathbb{E}_{\pi^{\mathrm{Trojan}}} \left[ \sum_{t=1}^{T} \gamma^{t-1} r_t^{(-)} \middle| s_1 \right]. \tag{2}$$

**Backdoor Mitigation.** Once a backdoor trigger has been identified, we aim to mitigate its effect during deployment by cleansing the Trojan agent's policy to avoid the trigger, thereby maximizing the overall expected return. To distinguish from the negated reward signal used in backdoor detection, we define the positive reward received by the Trojan agent as $r_t^{(+)} := \mathcal{R}(s_t, a_t^{\mathrm{Trojan}}, a_t^{\mathrm{adv}})$. Backdoor mitigation at deployment can be formulated as a maximization problem

$$\pi^{\mathrm{cleanse}} = \arg\max_{\pi} \ \mathbb{E}_{\pi} \left[ \sum_{t=0}^{H} \gamma^t r_t^{(+)} \middle| s_0, a_{1:T}^{\mathrm{adv}} \right], \ \text{where } \pi : \mathcal{S} \to \mathcal{A}^{\mathrm{Trojan}}, \tag{3}$$

where $\pi^{\mathrm{cleanse}}$ denotes the cleansed policy obtained by the defender via a test-time method, without retraining or fine-tuning the parameters of the Trojan policy.

**State Dependence.** Both the detection and mitigation objectives depend on the initial state of the trajectory. The detection objective in Eq. (2) seeks an adversarial action sequence that maximizes the cumulative adversarial reward starting from $s_1$, while $\pi^{\mathrm{cleanse}}$ corresponds to the optimal $H$-step policy for the Trojan agent starting from $s_0$. Importantly, this formulation does not restrict triggers to appear at the beginning of an episode. Any intermediate state can serve as a new root state for detection by the Markov property. Therefore, the framework captures triggers at arbitrary positions in a trajectory.

---

[4]Many safety-critical real-world RL applications, *e.g.*, robotics and autonomous driving, are developed with simulation environments for validation (Lee et al., 2020; Sinha et al., 2020; Wang et al., 2021a).

# 4 Methodology

We propose Plan2Cleanse, a RL backdoor detection and mitigation framework that leverages planning-based techniques to identify and neutralize hidden triggers in RL policies, with only black-box access to the Trojan policy. Unlike prior work that trains separate probing policies through iterative RL (Guo et al., 2023), our method directly searches for adversarial action sequences that activate backdoor behaviors without relying on gradient access or model retraining. Our methodology is composed of two complementary components: (1) a tree-search-based trigger discovery module (Section 4.1) that formulates detection as a trajectory-level planning problem, and (2) a lightweight mitigation module (Section 4.2) that performs a local online replanning strategy that uses MCTS to replace the Trojan policy's actions when adversary-triggered patterns are detected with alternatives during deployment. An overview of our Plan2Cleanse is shown in Figure 1.

## 4.1 Backdoor Detection via Tree Search

The goal of backdoor detection is to identify triggers that activate hidden malicious behaviors in a Trojan policy. This task is particularly challenging as Trojan behaviors are only activated under specific, sparse triggers, which can take the form of long action sequences or localized observation patches, making them difficult to discover through naive exploration.

**Key Idea: Recasting Backdoor Detection as a Planning Problem.** We unify backdoor detection across different attacks by recasting it as a planning problem. Instead of training an auxiliary model with gradient updates to identify triggers (Guo et al., 2023), we directly perform search over candidate inputs, which may take the form of temporally extended action sequences in continuous-control environments or localized perturbations in image-based domains. This perspective allows us to evaluate candidate triggers through environment interaction in a black-box manner, without requiring gradients or model parameters. Moreover, the search-based formulation can significantly improve sample efficiency by prioritizing exploration toward trajectories or regions that yield observable performance degradation, avoiding the local optima issues common in gradient-based approaches.

Based on this perspective, we employ MCTS (Kocsis & Szepesvári, 2006; Browne et al., 2012) to explore the space of adversarial inputs, guided by a scalar reward signal that reflects the degradation of the performance of the Trojan policy. This formulation prioritizes action sequences that induce performance degradation, enabling efficient trigger discovery in long-horizon settings where gradient-based methods often struggle.

**Tree-Search-Based Trigger Discovery.** In Section 3.4, we introduced the adversary's trigger sequence $a_{1:T}^{\mathrm{adv}}$ and the detection objective. In practice, backdoor activation is typically manifested through a noticeable degradation in the Trojan agent's task performance, such as the Humanoid agent falling, unexpected disconnection in O-RAN, or consistently suboptimal action choices in Atari. This observation naturally motivates the use of the Trojan agent's reward as a scalar signal for detection, since a reduction in reward indicates that the candidate adversarial sequence may have triggered malicious behaviour.

While we adopt the negated Trojan reward as the default detection score, this choice is not intrinsic to Plan2Cleanse. Our framework remains applicable under any reward formulation that reflects undesirable or malicious outcomes in the target system. This can be achieved by defining environment-specific detection rules that explicitly specify such outcomes (e.g., identifying falling events in the competitive Humanoid environment). These detection rules can be regarded as alternative reward signals. We define the evaluation score of a candidate adversarial sequence using the discounted cumulative negated reward:

$$Q(a_{1:T}^{\mathrm{adv}}) := \mathbb{E}_{\pi^{\mathrm{Trojan}}}\Big[\sum_{t=1}^{T} \gamma^{t-1} r_t^{(-)}\Big], \tag{4}$$

where $\gamma \in (0,1]$ is a discount factor.

**Continuous Action Selection via Voronoi-Based Sampling**. In continuous-control environments, the adversary operates over a high-dimensional continuous action space $\mathcal{A}^{\mathrm{adv}}$. To maintain search efficiency,

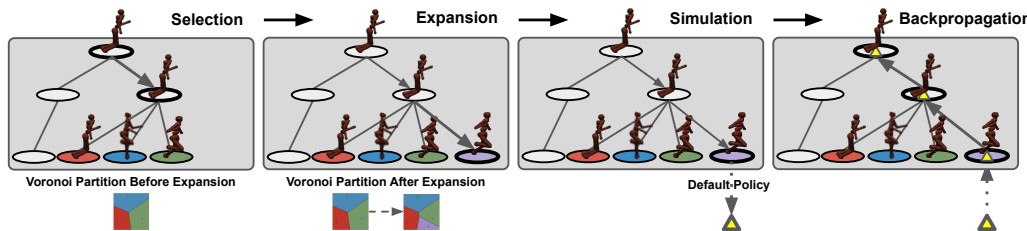

Figure 2: An illustration of Voronoi-based sampling in continuous action spaces. At a node corresponding to state $s_t$ before expansion, the previously expanded child actions $\mathcal{A}^{\text{adv}}_{\text{sampled}}(s_t)$ induce a Voronoi partition over the action space. Each child action defines a Voronoi region consisting of actions closer to it than to other sampled actions under the chosen metric. During expansion, new actions are sampled either globally or within the Voronoi region of the highest-valued child, progressively refining the partition. After expansion, promising regions become more finely subdivided, concentrating search density around high-value areas.

we incorporate a lightweight sampling strategy inspired by Voronoi Optimistic Optimization on Trees (Kim et al., 2020) to guide the selection of candidate adversarial actions.

At an MCTS node corresponding to state $s_t$, we maintain a node-local set of previously expanded adversarial actions $\mathcal{A}^{\text{adv}}_{\text{sampled}} = \{a_t^{(1)}, \ldots, a_t^{(n)}\}$, which contains only the child actions expanded at state $s_t$. These actions induce a Voronoi partition over the continuous action space $\mathcal{A}^{\text{adv}}$, where the cell associated with action $a_t^{(i)}$ is defined as

$$\mathcal{V}(a_t^{(i)}) = \{a \in \mathcal{A}^{\text{adv}} : \|a - a_t^{(i)}\| \leq \|a - a_t^{(j)}\|, \ \forall j \neq i\}. \tag{5}$$

As illustrated in Figure 2, each expanded action implicitly defines a region of influence in the continuous action space. This strategy balances global exploration with local refinement around promising regions. With probability $\omega$, the algorithm explores by sampling a new action uniformly over $\mathcal{A}^{\text{adv}}$; otherwise, it exploits by sampling within the Voronoi cell of the best-performing action to date:

$$\mathcal{V}^* := \arg \max_{\mathcal{V}(a_t^{(i)})} Q(a_{1:t}^{\text{adv}}). \tag{6}$$

This design enables both broad coverage of the action space and local refinement around high-scoring regions, supporting efficient trigger discovery under a limited interaction budget.

After simulating the environment forward under a sampled adversarial action and observing the resulting negated Trojan reward $r_t^{(-)}$, the estimated value of the state–action pair is updated via a max-backup rule:

$$\hat{Q}(s_t, a_t) \leftarrow r_t^{(-)} + \gamma \cdot \max_{a_{t+1}} \hat{Q}(s_{t+1}, a_{t+1}), \tag{7}$$

where the maximization is taken over all expanded child actions at state $s_{t+1}$. This recursive update propagates high adversarial scores upward through the tree, guiding future expansions toward action branches that exhibit stronger backdoor activation effects. The full procedure is provided in Algorithm 1.

**Remarks on the reward use for backdoor detection in Plan2Cleanse**. The core objective of Plan2Cleanse is not merely to flag arbitrary reward degradation, but to leverage MCTS to actively explore action sequences that induce systematically undesirable outcomes. In practice, these outcomes correspond to semantically meaningful failures (e.g., collapse, instability, or task violation), which typically manifest as consistent and substantial reward drops relative to the policy's expected trajectory. Moreover, even when primitive reward values are small in magnitude, the defender retains the flexibility to define undesirable system behaviors (e.g., instability, task failure, or safety violations), map them to safety- or task-level signals, and incorporate them into an augmented reward. Consequently, detection is not triggered by minor stochastic fluctuations, but by sequences that induce clear and semantically meaningful degradation relative to nominal policy behavior. This design ensures that the planning procedure focuses on structurally harmful behaviors rather than reacting to natural reward noise.

---

**Algorithm 1** Voronoi-Guided MCTS for Adversarial Action Selection

---

**Require:** Trojan policy $\pi^{\text{Trojan}}$, search budget $B$, exploration probability $\omega$, rollout depth $T$, adversary action space $\mathcal{A}^{\text{adv}}$

**Ensure:** Detection structure $\mathcal{D}$ with verified trigger sequences with Voronoi regions over $\mathcal{A}^{\text{adv}}$

1: Initialize tree root with current state $s_0$
2: **for** $b = 1, \ldots, B$ **do**
3:    $s_t \leftarrow \text{TreePolicy}(\text{root})$                        ▷ Traverse tree to a leaf node
4:    *// Select adversarial action via Voronoi-guided sampling*
5:    Retrieve sampled action set $\mathcal{A}^{\text{adv}}_{\text{sampled}} = \{a_t^{(1)}, \ldots, a_t^{(n)}\}$ at node $s_t$
6:    **if** $\mathcal{A}^{\text{adv}}_{\text{sampled}} = \emptyset \vee$ with probability $\omega$ **then**
7:       $a_t^{\text{adv}} \sim \text{Uniform}(\mathcal{A}^{\text{adv}})$                             ▷ Explore
8:    **else**
9:       Compute best Voronoi cell: $\mathcal{V}^* = \arg\max_{\mathcal{V}(a_t^{(i)})} Q(a_{1:t}^{\text{adv}})$
10:      $a_t^{\text{adv}} \sim \text{Uniform}(\mathcal{V}^*)$                               ▷ Exploit
11:    **end if**
12:    Add $a_t^{\text{adv}}$ to $\mathcal{A}^{\text{adv}}_{\text{sampled}}$
13:    *// Simulate and evaluate*
14:    Execute joint action $(a_t^{\text{Trojan}}, a_t^{\text{adv}})$ where $a_t^{\text{Trojan}} = \pi^{\text{Trojan}}(s_t)$; obtain $r_t^{(-)}$ and $s_{t+1}$
15:    $R \leftarrow \text{Rollout}(s_{t+1}, \pi^{\text{Trojan}}, T - t)$             ▷ Estimate return via rollout
16:    *// Backpropagate*
17:    Update $Q$-values along the path from $s_t$ to root using max-backup:
18:       $Q(a_{1:t}^{\text{adv}}) \leftarrow \max(Q(a_{1:t}^{\text{adv}}), r_t^{(-)} + \gamma \cdot R)$
19: **end for**
20: Construct $\mathcal{D}$ from highest-scoring action sequences in the tree
21: **return** $\mathcal{D}$

---

**Remarks on the possibility of missing rare triggers**. As MCTS cannot guarantee exhaustive exploration of the action space, consequently low-probability triggers may remain undiscovered. This limitation is inherent to most search-based exploration methods and is not unique to our method. However, unlike naive random search, MCTS mitigates this concern through reward-guided tree expansion, which prioritizes exploration toward action sequences that induce performance degradation, making it substantially more likely to discover low-probability triggers within a fixed interaction budget. This can be alleviated by repeating detection under multiple random seeds. Empirically, as shown in the sequel, in challenging environments such as minimal-responsiveness O-RAN, our method achieves substantially higher trigger detection success rates than baselines under comparable interaction budgets.

### 4.2 Backdoor Mitigation via Replanning

The objective of our mitigation module is to prevent the Trojan agent from exhibiting abnormal behaviors induced by backdoor triggers at test time, without retraining or fine-tuning the original Trojan policy. While our detection stage only identifies coarse quantized regions rather than exact trigger patterns, the effectiveness of mitigation nonetheless reflects the utility of the detection stage. In particular, successful mitigation indicates that the detected regions indeed capture the underlying triggers, since the replanning mechanism can only operate correctly when provided with meaningful detection results. Unlike detection, which identifies trigger sequences offline, mitigation operates online by dynamically replanning actions.

**Key Idea: Recasting Backdoor Mitigation as Preventive Replanning.** Trojan policies typically behave normally under benign conditions and begin to deviate only after trigger patterns gradually take effect. Our mitigation strategy aims to correct such deviations as early as possible, before the full trigger sequence is realized. While backdoor activation occurs immediately upon encountering the trigger, its impact on the trajectory typically accumulates over multiple steps, causing the agent to drift progressively away

from its benign behavior. Therefore, early correction steers the agent back toward normal behavior and prevents the deviation from compounding over time.

To implement this idea, we formulate mitigation as a preventive replanning problem at test time. When a suspicious adversary action is detected based on proximity to previously verified trigger actions, we initiate a localized search from the current state to identify an alternative. To allow flexible responses while preserving task performance, we use the Trojan policy as a sampling prior and inject stochastic noise during early search steps to encourage exploration. Among the explored actions, the one with the highest estimated return is selected to replace the Trojan policy's original action. This runtime mitigation module prevents backdoor activation without retraining or parameter modification, making it suitable for resource-limited or restricted-access settings. Instead of modifying the Trojan policy, we incorporate a lightweight replanning module that monitors execution and dynamically adjusts actions in response to observed adversarial patterns. This mechanism operates entirely in a black-box setting and aims to maintain overall task performance while countering adversarial behavior.

**Tree-Search Replanning for Efficient Backdoor Mitigation.** Building on the above intuition, our mitigation module requires a concrete mechanism to identify risky actions and propose alternative actions in real time. The key challenge is to efficiently decide when a candidate action is suspicious and how to generate a reliable replacement without modifying the underlying policy.

We leverage the detection results as a structural prior. The mitigation module uses the detection structure $\mathcal{D}$, constructed from verified trigger sequences identified during detection. Each node in $\mathcal{D}$ represents a state, and the associated Voronoi regions partition the action space around known adversarial actions. During test-time execution, given a state $s_t$ and adversary action $a_t^{\text{adv}}$, a fast geometric check determines whether $a_t^{\text{adv}}$ lies within a dangerous region. If so, a localized MCTS procedure is launched to replan the action and replace the potentially compromised output with an alternative.

---

**Algorithm 2** Backdoor Mitigation via MCTS Replanning

1: **Input:** State $s_t$, Detection Tree $\mathcal{D}$, Adversary action $a_t^{\text{adv}}$, Trojan policy $\pi^{\text{Trojan}}$, simulation budget $N$, horizon $H$
2: **Output:** Replanned action $a_t^{\text{Trojan}}$
3: **if** $a_t^{\text{adv}}$ in a dangerous region of $\mathcal{D}$ **then** ▷ Threat detected
4:     Initialize search tree $\mathcal{M}$ with root node $n_0 \leftarrow s_t$
5:     **for** $i = 1$ to $N$ **do**
6:         **for** $h = 1$ to $H$ **do**
7:             **if** $h < h_{\text{rollout}}$ **then**
8:                 $a_h \leftarrow \pi^{\text{Trojan}}(s_h) + \mathcal{N}(0, \sigma^2 I)$
9:             **else**
10:                 $a_h \leftarrow \pi^{\text{Trojan}}(s_h)$
11:             **end if**
12:         **end for**
13:         Backup rewards along the selected path
14:     **end for**
15:     $a_t^{\text{Trojan}} \leftarrow \arg\max_a \hat{Q}(n_0, a)$
16: **end if**
17: **return** $a_t^{\text{Trojan}}$

---

cedure is launched to replan the action and replace the potentially compromised output with an alternative.

We then perform localized tree-search replanning at test time. The replanning procedure constructs a search tree rooted at $s_t$, using the Trojan policy $\pi^{\text{Trojan}}$ as a sampling prior. Actions are selected as:

$$a^{\text{Trojan}} = \begin{cases} \pi^{\text{Trojan}}(s) + \mathcal{N}(0, \sigma^2 I), & \text{if } h < h_{\text{rollout}}, \\ \pi^{\text{Trojan}}(s), & \text{otherwise.} \end{cases} \tag{8}$$

When the depth $h$ reaches the rollout threshold $h_{\text{rollout}}$, the search switches to a deterministic rollout with the policy. The replanning return over the horizon $H$ is defined as:

$$R_{\text{replanning}} := \sum_{j=1}^{H} \gamma^{j-1} r_j^{(+)}. \tag{9}$$

This total return aggregates rewards from both phases of the simulation: stochastic exploration in the early steps and deterministic rollout in the latter part. This split between stochastic exploration and deterministic simulation mirrors the classical MCTS design. Restricting noise to the earlier depths ensures stability in action-value estimates, while still enabling local deviation from malicious trajectories. The replanning tree

backs up Q-values from rollout returns and selects the action at the root node with the highest estimated value. This replacement action is then executed in place of the potentially dangerous one.

The mitigation process is summarized in Algorithm 2, and the danger state labeling used to expand detection coverage is shown in Algorithm 4. A complete version of the replanning procedure is provided in Algorithm 5. During mitigation, the planner evaluates multiple candidate actions, including the original action generated by the Trojan model. The replanning module selects the action with the highest predicted return estimated through simulated rollouts. As the original action remains one of the evaluated candidates, the replanned action is selected based on the same simulated environment and reward computation, ensuring a fair comparison among candidates. Within the simulated horizon $H$, this procedure heuristically favors actions that maintain or improve short-horizon performance according to the planner's rollout estimates. Note that the adversary's actions are never directly altered, and the mitigation only modifies the Trojan policy's outputs upon detection of adversarial triggers.

Our method balances the use of learned policy actions with targeted divergence only when a threat is detected. This allows the agent to follow its original behavior under benign conditions while avoiding adversarial responses under backdoor activation. The result is a flexible and practical mitigation strategy applicable across environments, even in continuous action settings where fine-grained input filtering is infeasible.

**Remarks on long-term stability under the backdoor mitigation of Plan2Cleanse.** Plan2Cleanse is designed as a *selective, event-triggered intervention mechanism*, rather than a continuously active controller. Long-term stability is therefore preserved for two main reasons: (i) Replanning is invoked only when a trigger sequence is detected; under normal operation, the original policy executes unchanged. This avoids repeated overrides along benign trajectories and limits the risk of accumulated errors or policy drift. (ii) The replanning module does not update model parameters or permanently modify the learned policy. It operates purely at test time, producing localized corrective actions conditioned on detected triggers. Once the system returns to a normal state, control is handed back to the original policy. Consequently, there is no gradient update, memory accumulation, or internal-state modification that could compound over time, fundamentally differing from continual fine-tuning or adaptive schemes that may induce drift.

Empirically, as shown in the subsequent experiments across all evaluated domains (MuJoCo, O-RAN, and Atari), we observe no performance degradation over extended trajectories under repeated evaluations. Clean-task performance remains stable (e.g., Table 3 in Section 5.4), and no progressive instability appears during long rollouts. These results indicate that the lightweight, event-driven replanning mechanism does not introduce cumulative instability in practice.

In summary, Plan2Cleanse supports two practical deployment modes: (i) In the *pre-deployment auditing mode*, a larger search budget can be used to systematically probe different regions of the state space and stress-test the policy before release. (ii) In the *online safeguard mode*, the method can be applied selectively from the current state with a limited planning budget, triggering replanning-based mitigation only when suspicious behavior is detected. This allows the defender to balance computational cost and protection strength depending on operational constraints.

## 5 Evaluation

### 5.1 Setup

**Evaluation Domains.** We evaluate Plan2Cleanse across three RL domains: (1) **MuJoCo**: Competitive continuous-control tasks `run-to-goal-humans` and `run-to-goal-ants` (Bansal et al., 2018). (2) **O-RAN Wireless**: The communication-oriented `mobile-env` simulator (Schneider et al., 2022), featuring multiple UEs and base stations for cell association and power allocation. (3) **Atari**: Image-based control tasks where backdoors are embedded via small visual patches. Further experimental details are provided in Appendix B.1.

**Baselines.** We compare our defense method with baselines across two categories of backdoor attacks. For competitive multi-agent and mobile network environments (MuJoCo and `mobile-env`), we evaluate against three baselines: (1) *Normal*, which refers to a Trojan model operating without any trigger activation. This setting verifies that the Trojan model behaves normally under untriggered conditions and that any abnormal

behavior is indeed caused by the activation of the backdoor, rather than by spontaneous failures. It therefore serves as a false-positive control, ensuring that our detection pipeline does not falsely identify normal trajectories as triggers. (2) *Random*, which samples actions uniformly at random from the environment's action space to search for triggers. (3) *PolicyCleanse* (Guo et al., 2023), a recent backdoor detection framework.

For Atari, we compare against defenses for observation-level perturbation triggers: (1) *Provable Defense* (PD) (Bharti et al., 2022), which projects observations onto a safe subspace; (2) *BIRD* (Chen et al., 2023), which removes backdoor-sensitive neurons; (3) *Neural Cleanse* (Wang et al., 2019), which reverse-engineers and mitigates trigger patterns.

**Remark on Baselines.** SHINE (Yuan et al., 2024) is one recent backdoor defense method designed for poisoned RL environments and can serve as a good baseline. For reproduction, we utilized the official code of SHINE for Atari games and have exchanged several emails with the first author of SHINE about the detailed experimental configuration. Although we spent considerable effort trying to reproduce the results, unfortunately we were still not able to reach the performance reported in the paper. Despite this, we have tried our best and incorporated two recent and strong baselines, namely BIRD (Chen et al., 2023) and PD (Bharti et al., 2022), for evaluation in Atari.

**Evaluation Metrics and Trigger Criteria.** We report the *Trigger Detection Success Rate* (TDSR) (Guo et al., 2023), defined as the proportion of seeds for which at least one valid trigger is discovered. Each method is evaluated across 500 seeds, with up to 1000 episodes per seed. A discovered sequence is considered a valid trigger if it reliably induces the intended malicious behavior; full environment-specific acceptance rules are provided in Appendix B.1.

**Sanity Check on Benign Models.** Before evaluating the Trojan agents, we verify that Plan2Cleanse does not mistakenly identify any trigger on benign models. Running the full detection procedure on benign model across all environments produces no valid triggers, indicating that our method does not report backdoor triggers when none exist.

## 5.2 Results of O-RAN Simulator

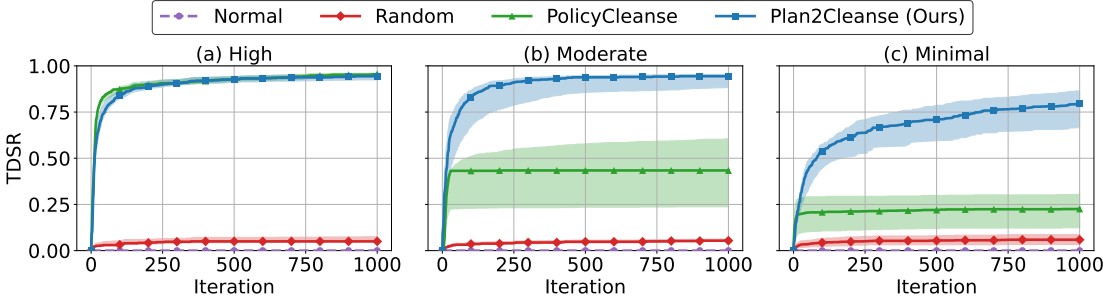

Figure 3: TDSR over training iterations in `mobile-env`. Solid lines show the median across seeds and shaded areas show the interquartile range. Plan2Cleanse sustains high detection performance even for minimally responsive Trojans, whereas PolicyCleanse drops notably as responsiveness decreases.

**Backdoor Detection in O-RAN Simulator.** We evaluate Plan2Cleanse in the `mobile-env` across three levels of Trojan responsiveness to triggers, which reflect how easily the backdoor behavior becomes active. The categories are: (1) **High responsiveness**, where the trigger gets activated in almost every evaluation; (2) **Moderate responsiveness**, where the trigger gets activated with medium frequency; (3) **Minimal responsiveness**, where the trigger is rarely activated. This categorization captures different levels of stealthiness and provides a clear basis for evaluating detection performance.

As shown in Figure 3, our Plan2Cleanse consistently outperforms all baselines across highly responsive, moderately responsive, and minimally responsive Trojan models. While PolicyCleanse performs comparably in the moderately responsive case, its success rate drops significantly when facing stealthier models. In

contrast, Plan2Cleanse maintains strong detection capability across all categories, achieving 0.735 even against minimally responsive Trojans. These findings highlight Plan2Cleanse's robustness across varying adversarial model behaviors, supporting its practical utility in realistic, communication-driven scenarios.

**Backdoor Mitigation in O-RAN Simulator.** We evaluate Plan2Cleanse's mitigation performance in the `mobile-env`, where benign UEs interact with a Trojan-infected base station controller. Figure 4 reports the average data rate (GB/s) under adversarial triggers across three Trojan responsiveness levels. Because mitigation is only activated upon trigger detection, responsiveness affects how often mitigation is applied rather than how it works. Plan2Cleanse consistently restores performance close to the benign policy (12.1 GB/s) across all settings, maintaining above 11.5 GB/s even in moderately and minimally responsive cases. While PolicyCleanse also improves performance over the Trojan baseline, its recovery remains lower than ours, and the gap widens as the Trojan becomes less responsive. These results show that Plan2Cleanse delivers more complete and stable recovery across varied Trojan behaviors.

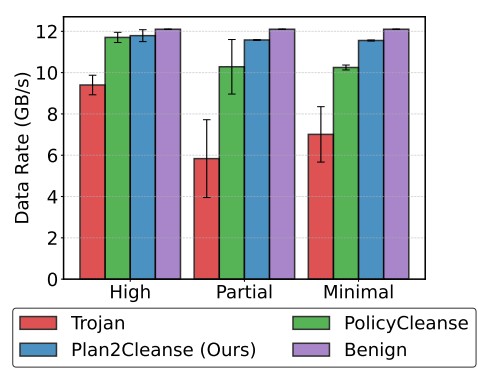

Figure 4: Average data rate (GB/s) under adversarial triggers in `mobile-env`.

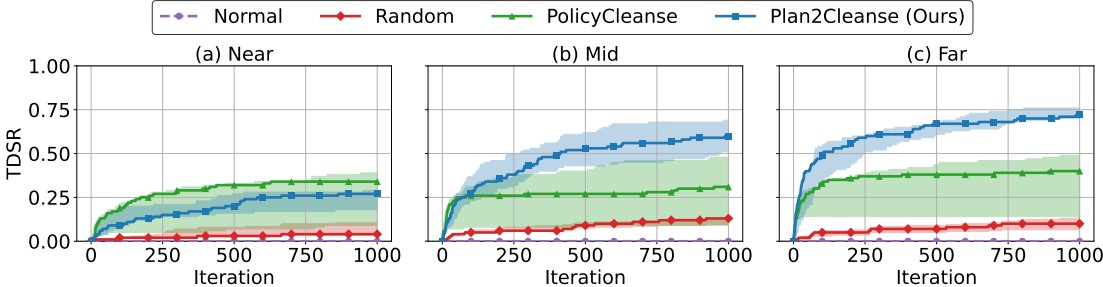

Figure 5: TDSR results under varying UE distances from the base station. Each plot reports the median and interquartile range computed over six Trojan models in the corresponding setting. Detection difficulty correlates with the strength of benign signals, with farther UEs exhibiting more detectable Trojan behaviors.

**Impact of UE Distance on Detection Performance.** We evaluate how the distance between the UEs and the base station influences Trojan detection, grouping scenarios by the theoretical maximum benign data rate: *Near* ($> 5$ GB/s), *Mid*, and *Far* ($< 0.5$ GB/s). As shown in Figure 5, Plan2Cleanse achieves the best detection performance in both Mid and Far settings, and is comparable to PolicyCleanse in Near. Detection is harder in Near because strong benign signals overshadow trigger effects, while increased distance amplifies Trojan-driven behaviors, improving detectability. In the Far setting, Plan2Cleanse reaches a 0.72 detection rate, outperforming PolicyCleanse at 0.40. This highlights that Plan2Cleanse remains reliable under low-signal conditions where Trojan behaviors are more dominant, making robust detection particularly valuable when UEs operate far from benign base stations.

Table 2: Final TDSR (*mean $\pm$ std*) at iteration 1000 across all environments. **Bold** and underlined results indicate best and second-best performance.

| Method | Ant | Humanoid | High | Moderate | Minimal |
|---|---|---|---|---|---|
| Uniform Random | $0.586 \pm 0.235$ | $0.030 \pm 0.021$ | $0.013 \pm 0.006$ | $0.020 \pm 0.006$ | $0.015 \pm 0.008$ |
| Normal Agent | $0.000 \pm 0.000$ | $0.000 \pm 0.000$ | $0.000 \pm 0.000$ | $0.000 \pm 0.000$ | $0.000 \pm 0.000$ |
| PolicyCleanse (Guo et al., 2023) | $0.595 \pm 0.175$ | $0.609 \pm 0.083$ | **$0.949 \pm 0.016$** | $0.484 \pm 0.208$ | $0.121 \pm 0.133$ |
| Plan2Cleanse (Ours) | **$0.946 \pm 0.032$** | **$0.997 \pm 0.010$** | $0.936 \pm 0.027$ | **$0.906 \pm 0.065$** | **$0.735 \pm 0.178$** |

## 5.3 Results of Competitive RL Environments

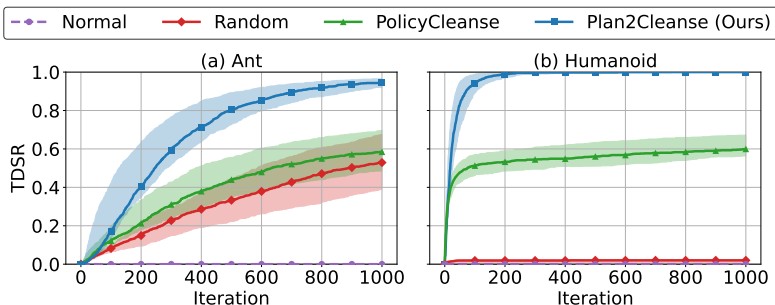

Figure 6: TDSR over training iterations in competitive RL environments. The solid lines show the median detection success rate across 500 randomized trials, with shaded regions indicating the interquartile range.

**Backdoor Detection in Competitive RL Environments.** In the `run-to-goal` environments, Figures 6(a) and 6(b) show that Plan2Cleanse consistently outperforms all baselines across both Humanoid and Ant. As summarized in Table 2, Plan2Cleanse reaches a final TDSR of 0.997 on Humanoid and 0.946 on Ant, surpassing PolicyCleanse by over 35 percentage points on average. Notably, we reproduce PolicyCleanse's Humanoid results under the same 1000-episode budget used by Plan2Cleanse and observe a detection rate of 60.9%, which aligns with the trend in their original curve; the higher 80% detection reported in the PolicyCleanse paper corresponds to a 3000-episode setting, indicating that our method achieves competitive results with substantially fewer interactions. Plan2Cleanse rapidly converges to near-perfect detection with high stability across seeds. The Normal baseline remains at zero throughout all iterations, as Trojan policies activate only under specific trigger patterns, making it a reliable lower bound on false positives. In Ant, the Random baseline occasionally reaches TDSR values comparable to PolicyCleanse, likely due to Ant's more constrained motion space, where random exploration has a non-trivial chance of unintentionally triggering Trojan behaviors. It is worth noting that the original PolicyCleanse paper did not evaluate random search as a standalone detection strategy on Trojan-infected agents. Our inclusion of the Random policy demonstrates the benefit of guided search over unguided exploration.

**Backdoor Mitigation in Competitive RL Environments.** We compare four agents under adversarial triggers: benign, Trojan without mitigation, Trojan with PolicyCleanse, and Trojan with Plan2Cleanse. As shown in Figure 7, Plan2Cleanse achieves the highest win rate in Humanoid with 53.3%, surpassing PolicyCleanse at 39.7%, the Trojan baseline at 34.7%, and even the benign agent at 47.0%. This demonstrates that local replanning not only removes backdoor effects but also stabilizes high-dimensional control. In Ant, PolicyCleanse reaches 40.3%, while Plan2Cleanse achieves 31.3%. Although slightly lower, our method still recovers a substantial portion of the lost performance, reflecting the challenge of mitigation in highly dynamic tasks where trigger behaviors are more subtle and distributed.

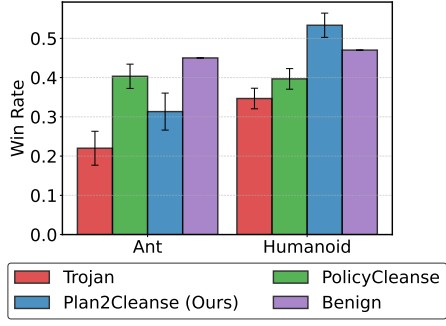

Figure 7: Win rate under adversarial triggers in Ant and Humanoid. Plan2Cleanse surpasses the benign policy in Humanoid.

## 5.4 Results of Atari Games

Table 3 summarizes the results for Pong and Breakout. Under poisoned inputs, our method Plan2Cleanse achieves competitive performance with PD and BIRD, while Neural Cleanse performs poorly. Under clean inputs, Plan2Cleanse preserves benign performance since no replanning is triggered without adversarial inputs. Plan2Cleanse can achieve performance comparable to BIRD and PD on poisoned Atari despite that it operates entirely at test time in a black box setting and does not require clean samples, unlike PD, BIRD, and Neural Cleanse which either require fine-tuning or need clean samples. In contrast, those baselines explicitly reconstruct the trigger pattern, while our Atari detection module only identifies quantized regions that influence the agent's action. Thus, it provides a location prior for mitigation rather than exact pattern

recovery, making a direct comparison on detection accuracy less meaningful. For fairness, we match the interaction budgets on Atari and keep sample usage within the same order of magnitude across methods. Additional studies on different attack scenarios in Atari are provided in Appendix C.

Table 3: Performance comparison of backdoor defenses in Atari games (Pong and Breakout) under both poisoned and clean environments. Results are reported as mean ± std scores. **Bold** results indicate the best performance. The results show that Plan2Cleanse can achieve comparable or better performance than the baselines, without any fine-tuning or clean data.

| Environment | Method | Category | Clean Data | Pong | Breakout |
|---|---|---|---|---|---|
| Poisoned | Trojan | - | - | $0.033 \pm 0.145$ | $16.25 \pm 0.63$ |
| | Neural Cleanse (Wang et al., 2019) | Fine-Tuning | Required | $-0.037 \pm 0.037$ | $6.45 \pm 0.28$ |
| | BIRD (Chen et al., 2023) | Fine-Tuning | Required | $0.960 \pm 0.016$ | $19.90 \pm 0.04$ |
| | PD (Bharti et al., 2022) | Test-Time | Required | $\mathbf{0.973 \pm 0.009}$ | $\mathbf{21.46 \pm 0.22}$ |
| | Plan2Cleanse (Ours) | Test-Time | Not Required | $0.950 \pm 0.033$ | $20.50 \pm 0.72$ |
| Clean | Trojan | - | - | $\mathbf{1.000 \pm 0.000}$ | $\mathbf{22.93 \pm 0.18}$ |
| | Neural Cleanse (Wang et al., 2019) | Fine-Tuning | Required | $0.867 \pm 0.009$ | $8.22 \pm 0.27$ |
| | BIRD (Chen et al., 2023) | Fine-Tuning | Required | $0.960 \pm 0.016$ | $19.90 \pm 0.04$ |
| | PD (Bharti et al., 2022) | Test-Time | Required | $0.973 \pm 0.009$ | $21.46 \pm 0.22$ |
| | Plan2Cleanse (Ours) | Test-Time | Not Required | $\mathbf{1.000 \pm 0.000}$ | $\mathbf{22.93 \pm 0.18}$ |

## 5.5 Computational Overhead

**Detection Overhead.** We compare the computational cost of our Plan2Cleanse with the baseline method PolicyCleanse in terms of the average number of environment interactions required to discover a valid backdoor trigger. In the Humanoid environment, our method identifies triggers with an average of 46 environment steps per trigger, versus 765 steps for PolicyCleanse. In the Ant environment, our method requires 384 steps on average, compared to 1296 for the baseline. These demonstrate the better efficiency of Plan2Cleanse than the baselines in backdoor detection.

**Mitigation Overhead.** The computational cost of mitigation depends on the environment. In the Humanoid environment, each mitigation step requires approximately $500 \times 4 = 2{,}000$ simulation steps, corresponding to 500 rollouts with a 4-step planning horizon. In the Ant environment, each mitigation step incurs a similar cost of $500 \times 4 = 2{,}000$ simulation steps. In the O-RAN simulator, which features lower-dimensional control, each mitigation step requires approximately $50 \times 1 = 50$ simulation steps, corresponding to 50 rollouts with a 1-step planning horizon. Since O-RAN systems emphasize near-real-time responsiveness, we also measure the wall-clock latency of our mitigation procedure. On a workstation equipped with an Intel Core i7-13700K CPU and an NVIDIA GeForce RTX 4070 Ti GPU, the mitigation latency averages $26.1 \pm 6.9$,ms per operation, which is well within the O-RAN near real-time control loop budget (10 ms–1s) (Raftopoulos et al., 2024). In the Atari environments, each mitigation step requires approximately $50 \times 20 = 1{,}000$ simulation steps in Pong (50 rollouts with a 20-step planning horizon) and $30 \times 20 = 600$ simulation steps in Breakout (30 rollouts with a 20-step planning horizon).

For a fair comparison, we let fine-tuning baselines use a similar order of magnitude of interactions. PolicyCleanse performs fine-tuning on 50,000 state-action pairs in MuJoCo and 1,000 state-action pairs in O-RAN, which are comparable to our simulation budgets. BIRD uses about 30,000 environment steps for Pong and 50,000 for Breakout, while Provable Defense (PD) requires 12,000 clean environment steps for Atari.

## 5.6 Sensitivity Analysis of Key Hyperparameters

We further evaluate the sensitivity of both $H$ and $h_{\text{rollout}}$, which are the two major hyperparameters in Algorithm 2. Figure 8 demonstrates the sensitivity of Plan2Cleanse to the planning horizon and the rollout threshold in Humanoid. The planning horizon $H$ determines the depth of tree search, and our experiments show that increasing $H$ beyond a small value brings little additional benefit. The performance improves quickly as $H$ increases, stabilizing once $H \geq 4$, which shows that the shallow lookahead is sufficient under

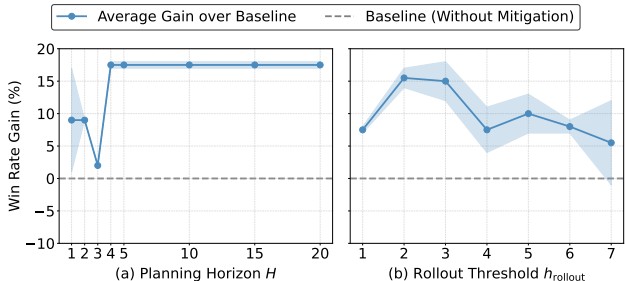

Figure 8: Effect of planning horizon $H$ and rollout threshold $h_{\text{rollout}}$ on win rate gain (%) in the Humanoid environment.

fixed budgets. On the other hand, the rollout threshold $h_{\text{rollout}}$ controls stochastic exploration during search: small values restrict exploration and risk imitating the Trojan policy, while large values inject excessive noise and lead to unstable value estimates; moderate settings (*i.e.*, around 3 to 5) provide the best balance. Similar sensitivity curves for *mobile-env*, MuJoCo, and Atari are provided in Figures 15, 16, and 17 in Appendix F.1.

Notably, Plan2Cleanse is also insensitive to the choice of detection depth $T$, as strong detection performance is achieved with small $T$ and remains stable across a broad range of values, as shown in Figures 12 and 13 in Appendix F.3.

## 6 Conclusion

In this work, we propose Plan2Cleanse, a novel framework that employs MCTS to efficiently detect and mitigate backdoor attacks in RL without requiring model or policy retraining. Our approach offers significant advantages over existing methods by formulating backdoor detection as a trajectory-level planning problem and introducing a lightweight mitigation strategy that neutralizes malicious behaviors through local replanning. Through extensive experiments in competitive multi-agent environments, simulated O-RAN systems, and Atari games, we substantially outperform state-of-the-art methods, consistently detecting more backdoor triggers while requiring fewer interactions with the system. Our findings demonstrate that even highly stealthy backdoors can be discovered and neutralized in practical scenarios like wireless network management, highlighting the critical need for robust detection and mitigation strategies to secure RL systems against backdoor threats in real-world deployment. In principle, the replanning returns could be used as training signals to iteratively refine the Trojan policy, similar to planning-based optimization methods such as AlphaZero. However, since Plan2Cleanse operates in a strict black-box setting without access to model parameters, incorporating gradient-based updates falls outside our scope and is left for an promising future work in white-box scenarios.

### Acknowledgments

This research was partially supported by the National Science and Technology Council (NSTC) of Taiwan under Grant Numbers 114-2628-E-A49-002 and 114-2634-F-A49-002-MBK. This work was also partially supported by the Center for Intelligent Team Robotics and Human-Robot Collaboration under the "Top Research Centers in Taiwan Key Fields Program" of the Ministry of Education (MOE), Taiwan. The authors also thank the National Center for High-performance Computing (NCHC) for providing computational and storage resources. The authors thank the anonymous reviewers for the constructive comments, technical questions, and invaluable suggestions for presentation that led to an improved text.

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

# A    Threat-Model Instantiations in Realistic RL Deployments

To further substantiate the threat model in Section 3.3, we describe how it is realized in the three representative domains considered in our experiments.

- **O-RAN Wireless Networks.** In this setting, the target policy operates as a control module within a simulated wireless network, and it receives input features derived from the measurements of a user equipment (UE) and produces control decisions for cell association and power allocation (Liyanage et al., 2023). The adversary is embedded in one UE, which behaves maliciously during deployment by emitting carefully crafted sequences of physical-layer measurements. These sequences are intended to activate the backdoor and induce the controller to deviate from its expected behavior. The defender can simulate various network conditions and configure UE behavior to probe for such triggers, but cannot access or modify the internal logic of the target policy.

- **Competitive Robot Control.** In this setting, the target agent policy operates within a competitive control environment (Gleave et al., 2020; Wang et al., 2021b). The adversary is implemented as an opponent agent that executes specific action sequences that can trigger abnormal behavior on the target. The defender interacts with the system by simulating episodes with different opponent behaviors, which may be scripted, stochastic, or search-based, but cannot access or modify the internal logic of the target policy.

- **Atari Image-Based Control.** The target policy operates in an image-based Atari game environment. The adversary exposes observation-level triggers as localized patches at deployment to induce abnormal actions (Kiourti et al., 2020). The defender has black-box access during deployment, and hence it observes trajectories and can override the agent's actions (*e.g.*, through online replanning) but cannot inspect or modify policy parameters.

# B    Detailed Experimental Configurations

## B.1    Key configurations

**Trojan Model Construction.**    We construct Trojan agents across all environments following standard backdoor setups. For MuJoCo (Ant and Humanoid) and `mobile-env`, we adopt the BackdooRL framework (Wang et al., 2021b) and train 10 distinct Trojan policies per environment by mixing 2000 benign episodes with 500 poisoned episodes. Triggers are injected as action sequences that cause abnormal behaviors, including manually designed motion patterns (*e.g.*, bending a specific limb in Humanoid or Ant) and randomly sampled action subsequences for greater diversity. In `mobile-env`, triggers are realized through injected physical-layer signal patterns that reliably activate malicious scheduling behaviors. For Atari (Pong and Breakout), we follow the SleeperNets protocol (Rathbun et al., 2024) to implant patch-style triggers and retrain Trojan agents accordingly. These Trojan models form the evaluation testbed for both detection and mitigation.

**Trigger Criteria.**    To evaluate whether a discovered sequence constitutes a successful trigger, we define environment-specific acceptance criteria:

- **Ant**: A trigger is accepted if it causes a statistically significant drop in the victim agent's return. We follow the prior work (Wang et al., 2020; Guo et al., 2022) and apply an anomaly detection procedure based on the Median Absolute Deviation (MAD). Let $r_{\text{sum}}$ denote the negated cumulative reward of the replayed candidate sequence, and let $r_{\text{ref}}$ be a reference distribution obtained from 500 random action sequences. We compute the anomaly index as:

$$\text{Anomaly Index}(r_{\text{sum}}) := \frac{r_{\text{sum}} - \text{Median}(r_{\text{ref}})}{C \cdot \text{Median}(|r_{\text{ref}} - \text{Median}(r_{\text{ref}})|)},$$

  where $C = 1.4826$ is a scaling constant assuming Gaussian residuals. A trigger is accepted if the index exceeds the threshold:

$$\text{threshold} := 4 \cdot C \cdot \text{MAD} + \text{Median}(r_{\text{ref}}),$$

corresponding to a 4-MAD outlier detection rule.

- **Humanoid**: A trigger is considered valid if it causes the agent to fall, operationalized as a sudden drop in center-of-mass height during evaluation.

- **Mobile-env**: A trigger is accepted if the adversarial UE disrupts a benign UE's connection despite prior signal stability.

- **Pong and Breakout**: Accept a candidate trigger if (i) there exist frames where inpainting exactly one grid cell flips the policy's action (discard frames with zero or multiple flips), and (ii) the vote map over frames has a unique argmax; we report the average pixels of that cell as the patch.

Note that the principle of our detection framework remains environment-agnostic, even though the trigger acceptance criteria are task-specific. It is natural to use the return as the performance indicator when the environment provides an informative reward signal. For environments where the reward does not directly reflect the undesirable behavior, practitioners may define environment-specific criteria that formalize the detection of such events.

These task-specific criteria can be interpreted as practitioner-defined reward functions. For example, the MAD-based thresholding in Ant instantiates a reward-driven detection rule based on statistically significant decreases in return and can be readily adapted to other RL domains with meaningful reward signals. In contrast, the fall detection rule in Humanoid represents a domain-specific event-based criterion and requires additional domain knowledge for adaptation.

**Detection and Mitigation Parameters.** We use Gaussian exploration noise with a standard deviation of $\sigma = 0.1$, and set the Voronoi Optimistic Optimization (VOO) sampling radius to 0.1.

The *rollout threshold* $h_{\mathrm{rollout}}$ determines the depth at which the planner transitions from stochastic exploration to deterministic rollout, balancing exploration diversity with stable value estimation. Environment-specific configurations for the budget $N$, rollout threshold $h_{\mathrm{rollout}}$, and planning horizon $H$ are summarized in Table 4.

Table 4: Environment-specific hyperparameters for Plan2Cleanse detection and mitigation.

| Parameter | Ant | Humanoid | Mobile-env | Pong | Breakout |
|---|---|---|---|---|---|
| Detection Depth $T$ | 60 | 10 | 10 | 1 | 1 |
| Mitigatoin Budget $N$ | 500 | 500 | 10 | 30 | 50 |
| Rollout Threshold $h_{\mathrm{rollout}}$ | 3 | 3 | 5 | 1 | 1 |
| Planning Horizon $H$ | 5 | 5 | 5 | 20 | 20 |

**Baseline Reproduction.** For baseline reproduction, we matched the environment step magnitudes to our method and configured poisoned environments with the same poisoning rates as used in evaluation, namely 0.25 in Pong and 0.2 in Breakout. This setup was intended to align with SHINE (Yuan et al., 2024). However, despite following the default configurations and further attempting to tune hyperparameters, we were unable to reproduce the reported improvements of SHINE in our setting. We also contacted the authors and exchanged emails to confirm the experimental details, but the reproduced performance remained inconsistent with their published results.

### B.2 Additional Analysis and Ablations

**Action Perturbation Strategies in MuJoCo Mitigation.** Our mitigation approach requires sampling alternative actions around the Trojan policy to avoid dangerous behaviors. We evaluate three perturbation strategies: Gaussian noise ($\mathcal{N}(0, 0.1^2 I)$) with standard deviation 0.1, uniform sampling with VOO, and OU noise which introduces temporally correlated deviations. Results in Figure 9 show that Gaussian noise performs best in Ant environments with low variance, while OU noise achieves superior performance in the high-dimensional Humanoid setting. This suggests that temporally correlated noise is more effective for complex control tasks, while simple Gaussian perturbation suffices for lower-dimensional environments.

**More Details on Backdoor Detection in Atari.** In addition to the criteria specified in Section 5.1, we partition each Atari frame into $12 \times 12$ grids, yielding 49 ($7 \times 7$) candidate patches. A trigger is accepted if inpainting a unique patch consistently flips the chosen action. To demonstrate the effectiveness of this quantized patch detection approach, we report the voting ratios of trigger regions across different trigger patterns. For each poisoned agent, we collect 1000 frames under a poisoning rate of 0.1 with trigger size set to $4 \times 4$. A frame is considered valid if exactly one patch flip is observed, and we record the patch location that received the vote. The trigger region voting ratio is computed as:

$$\text{Vote Ratio} = \frac{\text{Trigger Region Votes}}{\text{Total Valid Votes}}.$$

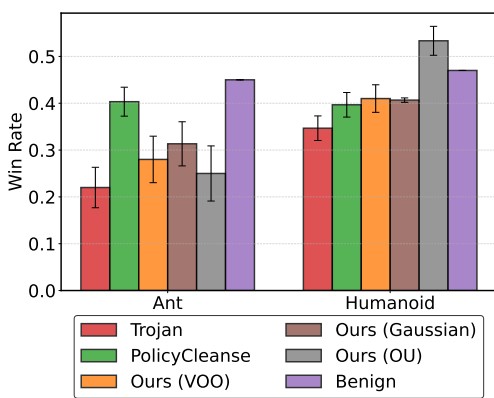

Figure 9: Win rates under different perturbation strategies in the Ant and Humanoid environments. Each bar shows the average performance across three Trojan models.

The results show strong detection accuracy: Equal (44/62, ≈71.0%), Cross (35/50, 70.0%), Checkerboard (60/64, 93.8%), and Square (14/15, 93.3%). These high voting ratios demonstrate that our quantized detection method effectively identifies the correct trigger regions across different patch patterns. For the benign model with trigger injection, we evaluate 1000 frames under the same detection setup. Among them, only 18 frames are valid (*i.e.*, those producing a single patch flip). The highest voting ratio within valid frames is 9/18 (50%), which does not exceed half of the total votes. When normalized over all evaluated frames, the trigger region ratio becomes 9/1000 (0.9%), which is below 1%. Therefore, we consider that no consistent trigger pattern is detected in the benign model.

**Mitigation Environment Setting of Atari.** To match the reported results of SHINE, we terminate each Pong episode once a non-zero reward is obtained. For Breakout, we truncate the episode length at 550 time steps. We also set the poisoning rates to 0.25 in Pong and 0.20 in Breakout, and use Trojan models with a $3 \times 3$ square trigger in Table 3.

## C Various Attack Scenarios in Atari Games

Table 5: Performance comparison under poisoned and clean environments for $4 \times 4$ patterns. Results are mean ± std.

| Environment | Method | Square | Equal | Cross | Checkerboard |
|---|---|---|---|---|---|
| Poisoned | Trojan | $0.033 \pm 0.145$ | $-0.127 \pm 0.064$ | $-0.147 \pm 0.170$ | $0.053 \pm 0.189$ |
| | Plan2Cleanse (Ours) | $0.950 \pm 0.014$ | $0.973 \pm 0.012$ | $0.787 \pm 0.151$ | $0.880 \pm 0.060$ |
| Clean | Trojan | $1.000 \pm 0.000$ | $1.000 \pm 0.000$ | $0.940 \pm 0.020$ | $1.000 \pm 0.000$ |
| | Plan2Cleanse (Ours) | $1.000 \pm 0.000$ | $1.000 \pm 0.000$ | $0.940 \pm 0.020$ | $1.000 \pm 0.000$ |

Table 6: Performance of the original poisoned agent and our method with square block patterns.

| Agent | 3×3 | 4×4 | 5×5 |
|---|---|---|---|
| Trojan | $-0.067 \pm 0.046$ | $0.033 \pm 0.145$ | $-0.093 \pm 0.070$ |
| Plan2Cleanse (Ours) | $0.993 \pm 0.012$ | $0.950 \pm 0.014$ | $0.753 \pm 0.050$ |

To further validate the robustness of our approach, we conducted additional experiments under diverse attack settings. Table 5 reports results across four trigger patterns (*Square*, *Equal*, *Cross*, *Checkerboard*, as shown in Figure 10). In poisoned environments, the original model suffers severe performance degradation, with accuracy dropping close to zero or even negative values, while our method consistently maintains performance close to 1.0. In clean environments, both methods achieve optimal performance, confirming that our defense



Figure 10: Different trigger patterns: (a) Square, (b) Equal, (c) Cross, and (d) Checkerboard.

does not compromise normal task execution. Table 6 further evaluates agents of different sizes ($3 \times 3$, $4 \times 4$, $5 \times 5$). The original agents exhibit substantial vulnerability under poisoning, whereas agents retrained with our method remain highly robust across all configurations, maintaining strong performance even as the environment scale increases. These results collectively demonstrate the effectiveness and stability of our approach across a wide range of attack scenarios.

## D   Additional Related Work on Adversarial Threats in RL-based O-RAN Applications

O-RAN represents a paradigm shift in wireless network management, promoting openness, flexibility, and interoperability by leveraging artificial intelligence (AI) and machine learning (ML) solutions. In particular, RL-based applications, often implemented as real-time xApps or non-real-time rApps within the RAN Intelligent Controller (RIC), are deployed to optimize crucial network functions such as resource allocation, traffic scheduling, handover management, and anomaly detection. RL-driven xApps (Mismar et al., 2020; Tang et al., 2023) have been investigated in both theoretical and experimental studies, showing their potential to dynamically optimize radio resource allocation and improve network performance in terms of throughput, latency, and reliability. However, the increasing reliance on ML and RL within O-RAN introduces significant vulnerabilities to adversarial threats. Prior work (Chiejina et al., 2024) has demonstrated that ML-based components deployed in the near-real-time RIC are susceptible to adversarial attacks capable of degrading network performance and manipulating control decisions. These attacks exploit the openness of O-RAN and the shared access to system data, enabling malicious xApps to inject carefully crafted perturbations that mislead legitimate applications. In particular, interference classification xApps (Sapavath et al., 2023) experience a marked decline in prediction accuracy under adversarial manipulation, leading to measurable deterioration in system throughput and capacity. The modular and decentralized nature of O-RAN (Farooq et al., 2019) further amplifies the risk, as compromised agents may tamper with shared observations or disrupt the behavior of co-located services. Addressing such threats demands a rigorous analysis of attack surfaces unique to ML-based control architectures and the design of robust, context-aware defense mechanisms.

## E   Visualization and Quantitative Analysis of Trigger Similarity

To further verify that the recovered triggers correspond to the true backdoor mechanisms rather than arbitrary adversarial trajectories, we provide both qualitative and quantitative evidence. Here, **True** denotes the ground-truth backdoor trigger used to implant malicious behavior, **Found** refers to the trigger recovered by Plan2Cleanse, and **Benign** represents trajectories sampled from a clean model without any backdoor.

**t-SNE Visualization.** Figure 11 visualizes the t-SNE embeddings of trigger sequences in both the O-RAN and continuous-control environments. Recovered triggers (Plan2Cleanse) cluster closely with the true triggers, while remaining clearly separated from benign trajectories. This visual evidence indicates that our method successfully rediscovers the underlying backdoor pattern rather than random failure trajectories.

**Quantitative Similarity.** We further compute the distance between the recovered and true triggers using multiple trajectory metrics (L1, L2, and DTW). For each environment, one ground-truth trigger sequence is used as **True**, while 100 recovered and 100 benign trajectories are sampled for averaging. Table 7 reports

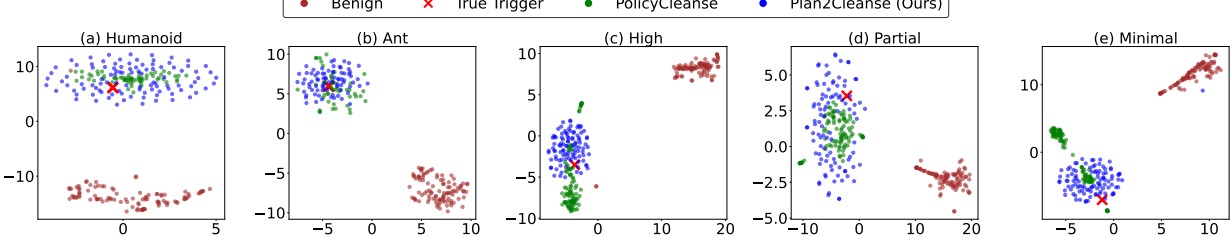

Figure 11: t-SNE visualization of trigger action sequences across five settings: Ant, Humanoid, and three mobile network variants with different Trojan responsiveness levels. Each point denotes a 10-step action sequence projected into 2D space, with colors representing different trigger sources. The true trigger is marked with a × symbol.

results across all five environments. In every case, the True–Found distance remains consistently smaller than both True–Benign and Found–Benign, demonstrating strong geometric alignment between the recovered and ground-truth triggers.

Together, these visual and numerical results confirm that Plan2Cleanse recovers triggers that closely match the true backdoor patterns and remain clearly separated from benign or adversarial trajectories.

Table 7: Trigger similarity across environments and metrics. Each cell reports mean ± std of distances between True, Found, and Benign triggers.

| Metric | Pair | Ant | Humanoid | High | Partial | Minimal |
|--------|------|-----|----------|------|---------|---------|
| L1 | True–Found | $4.515 \pm 0.299$ | $8.857 \pm 0.404$ | $1.169 \pm 0.188$ | $1.036 \pm 0.130$ | $1.191 \pm 0.177$ |
| | True–Benign | $5.621 \pm 0.365$ | $12.745 \pm 1.049$ | $1.620 \pm 0.224$ | $1.375 \pm 0.290$ | $1.592 \pm 0.230$ |
| | Found–Benign | $5.961 \pm 0.461$ | $14.772 \pm 0.945$ | $1.529 \pm 0.274$ | $1.531 \pm 0.342$ | $1.512 \pm 0.311$ |
| L2 | True–Found | $1.884 \pm 0.113$ | $2.512 \pm 0.106$ | $0.768 \pm 0.108$ | $0.675 \pm 0.074$ | $0.775 \pm 0.099$ |
| | True–Benign | $2.384 \pm 0.134$ | $3.703 \pm 0.248$ | $1.144 \pm 0.157$ | $0.959 \pm 0.161$ | $1.114 \pm 0.165$ |
| | Found–Benign | $2.537 \pm 0.182$ | $4.367 \pm 0.245$ | $1.013 \pm 0.187$ | $1.003 \pm 0.232$ | $1.009 \pm 0.220$ |
| DTW | True–Found | $1.884 \pm 0.113$ | $2.512 \pm 0.106$ | $0.650 \pm 0.109$ | $0.638 \pm 0.112$ | $0.717 \pm 0.126$ |
| | True–Benign | $2.384 \pm 0.134$ | $3.703 \pm 0.248$ | $1.144 \pm 0.157$ | $0.959 \pm 0.161$ | $1.114 \pm 0.165$ |
| | Found–Benign | $2.452 \pm 0.202$ | $4.363 \pm 0.248$ | $1.012 \pm 0.188$ | $1.002 \pm 0.233$ | $1.007 \pm 0.221$ |

# F   Extended Sensitivity Analysis Across Environments

This section provides additional sensitivity analyses for both the detection and mitigation components of Plan2Cleanse across different environments.

## F.1   Detection Depth Sensitivity

We study how the detection depth $T$ affects Trojan detection across *mobile-env* and MuJoCo. As shown in Figure 12 and Figure 13, increasing $T$ improves performance up to a moderate range, after which detection stabilizes. While the optimal $T$ differs across environments (*e.g.*, smaller $T$ suffices in *mobile-env*), detection remains strong over a broad range of $T$, showing that Plan2Cleanse does not rely on precise tuning of this parameter. Shaded areas represent the standard deviation across Trojan models, with each point showing the mean performance over different Trojan models.

## F.2   Exploration Probability Sensitivity

We further examine the role of the exploration probability $\omega$ in MuJoCo. As shown in Figure 14, detection performance remains consistent across a wide range of $\omega$ values, suggesting that the search procedure is

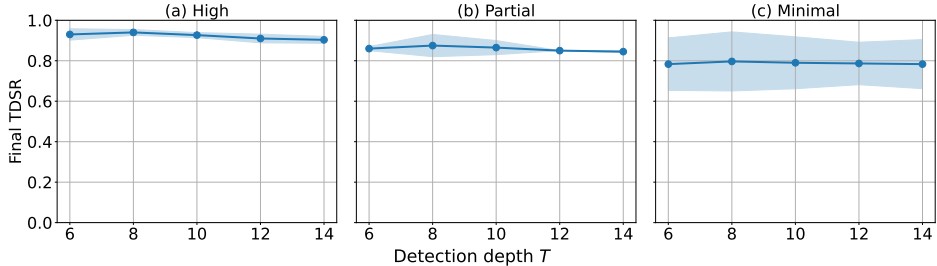

Figure 12: Effect of detection depth $T$ on final TDSR in *mobile-env* under High, Partial, and Minimal Trojan responsiveness.

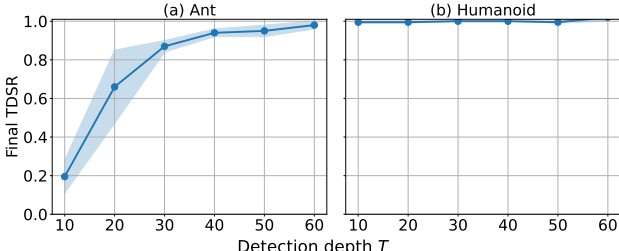

Figure 13: Effect of detection depth $T$ on final TDSR for Ant and Humanoid.

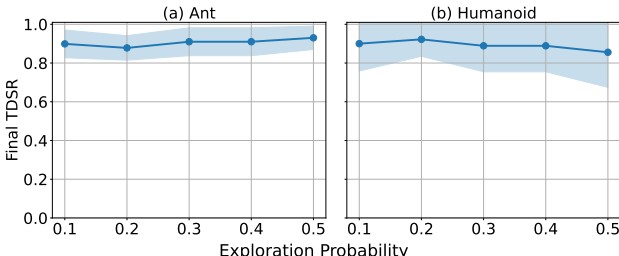

Figure 14: Effect of exploration probability $\omega$ on final TDSR for Ant and Humanoid.

robust to the exploration–exploitation trade-off. The shaded regions indicate the standard deviation across Trojan models, and each point corresponds to the mean performance aggregated over different Trojan models.

## F.3 Mitigation Hyperparameter Sensitivity

To complement the Ant results presented in Section 5.6, we further provide the sensitivity analysis of the mitigation hyperparameters across the Humanoid and mobile-env environments. For Humanoid, the overall trend remains consistent with Ant, where performance improves rapidly with increasing planning horizon $H$ and saturates when $H \geq 4$, indicating that shallow lookahead is sufficient for stable mitigation (Figure 8).

These additional results confirm that the trends observed in Ant generalize across high-dimensional locomotion, communication-control environments, and visual Atari domains, further validating the robustness of our mitigation design. For the *mobile-env*, we evaluate three categories of Trojan responsiveness (High, Partial, and Minimal), and use a unified planning horizon $H = h_{\mathrm{rollout}}$ in this environment. Only the continuous-control locomotion tasks (Ant and Humanoid) require a separate $h_{\mathrm{rollout}}$ due to their high-dimensional action spaces. Figure 16 shows that our method remains stable across different horizons, and even short planning horizons are sufficient to achieve effective mitigation.

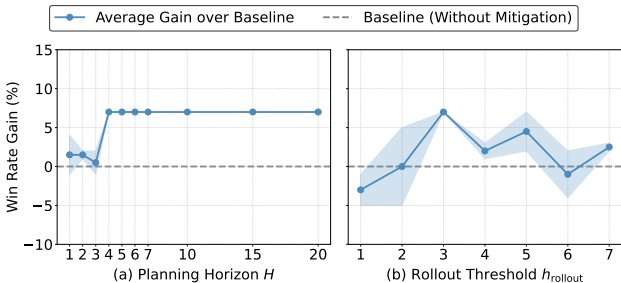

Figure 15: Effect of planning horizon $H$ and threshold $h_{\text{rollout}}$ on win rate gain (%) in the Ant environment.

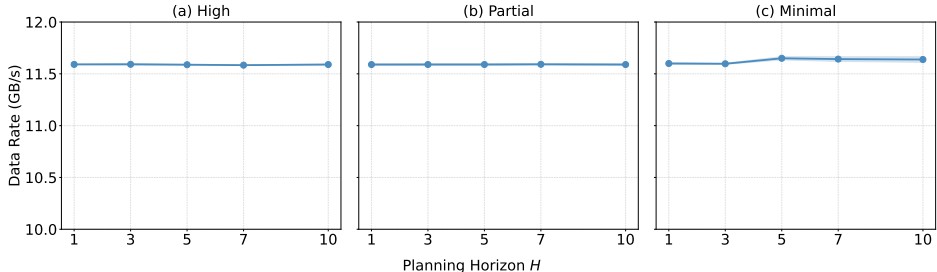

Figure 16: Effect of joint planning horizon $H = h_{\text{rollout}}$ on data rate (GB/s) in the *mobile-env*.

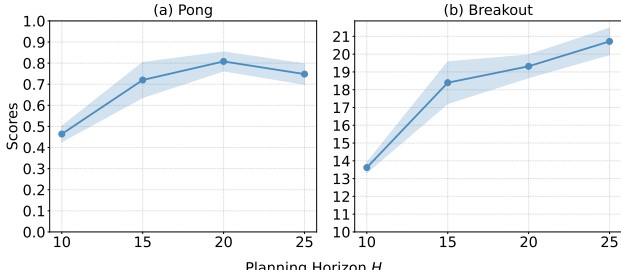

Figure 17: Effect of planning horizon $H$ on mitigation performance in Atari: (a) Pong and (b) Breakout.

# G   Ablation Study

This section provides additional analyses under noisy-reward and sparse-reward settings in MuJoCo to further evaluate the robustness of Plan2Cleanse.

## G.1   Noisy-Reward Setting

To assess robustness against noisy reward, we inject zero-mean Gaussian noise $\mathcal{N}(0, \sigma^2)$ into the reward at each time step. This introduces the noise standard deviation $\sigma$ as an additional hyperparameter. All other hyperparameters remain identical to those used in the main experiments (Table 4). We report the final TDSR at varying noise levels in Figure 18, averaged over 30 random seeds.

The results show that TDSR remains largely stable, indicating minimal sensitivity to reward perturbations. Moreover, even under high-variance noise, Plan2Cleanse still produces zero false positives over 100,000 evaluation episodes. These findings demonstrate that Plan2Cleanse remains reliable even when reward signals are inaccurate.

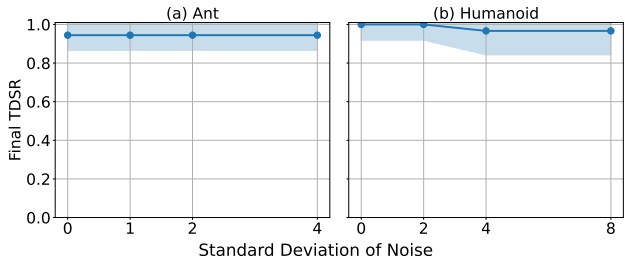

Figure 18: Effect of noise standard deviation $\sigma$ on final TDSR for Ant and Humanoid.

## G.2 Sparse-Reward Setting

We further evaluate Plan2Cleanse under sparse rewards. Specifically, the environment reward is set to zero at every time step except at episode termination: the agent receives $+1$ upon winning and $-1$ upon losing. This modification introduces no additional hyperparameters. All other configurations follow those of the main experiment (Table 4). Figure 19 reports the TDSR across iterations under sparse rewards, averaged over 30 random seeds.

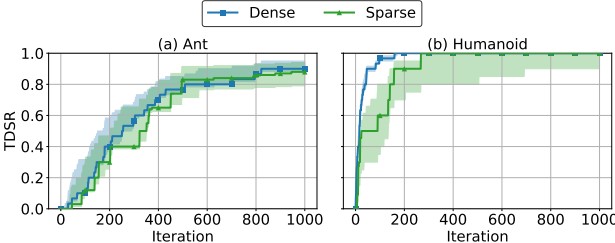

Figure 19: TDSR over training iterations in Ant and Humanoid. Solid lines show the median across seeds, and shaded areas show the interquartile range.

We observe that the final TDSR remains close to 1, comparable to the original dense-reward setting. This indicates that Plan2Cleanse can reliably detect backdoor triggers even with sparse reward signals.

# H More Details on Our Algorithms

## H.1 Adaptation to Perturbation-Based Triggers

The tree-search formulation in Section 4.1 targets multi-step, temporally extended triggers. In image-based domains such as Atari, backdoors are often activated by a single localized perturbation on the observation rather than by a temporal sequence of actions. To accommodate this setting within the same planning framework, we adapt the search to operate over spatial perturbations on a single frame.

When a trigger patch is present on the frame, the Trojan policy tends to choose a poisoned action, either targeted or untargeted. If we remove that local evidence by inpainting the true trigger region the policy reverts to its benign action, while masking any other region leaves the decision unchanged. We exploit this locality within the same planning framework by viewing screening as a depth-1 tree search. For a frame $s_t$, we define a discrete set of screening actions $\mathcal{A}_{\text{scr}}^{\text{adv}} = \{\text{mask}(i,j)\}_{i \leq L,\, j \leq W}$ by quantizing the image into an $L \times W$ grid; applying $\text{mask}(i,j)$ inpaints cell $(i,j)$ to produce a modified observation $\tilde{s}_t^{(i,j)}$. We first query $\pi^{\text{Trojan}}$ on $s_t$ to obtain the baseline action, then query on each $\tilde{s}_t^{(i,j)}$ and record the set of cells that change the chosen action. We keep the frame only if exactly one cell flips the action and discard frames with zero or multiple flips. For a kept frame we add one vote to that unique coordinate and accumulate the raw pixels of that cell for averaging. Repeating this under a fixed interaction budget yields a vote heatmap $C \in \mathbb{N}^{H \times W}$ whose argmax serves as a location prior, and the averaged patch provides a qualitative check. This screening

is black-box and training-free, does not attempt trigger reconstruction, and the resulting prior is used to gate online replanning in the mitigation stage. The full procedure is provided in Algorithm 3.

---

**Algorithm 3** Atari patchwise screening

---
1: **Input:** grid size $H \times W$, inpaint radius $r$, interaction budget $B$, policy $\pi$
2: **Output:** location prior $(i^\star, j^\star)$, vote heatmap $C \in \mathbb{N}^{H \times W}$, average patch $\bar{P}$
3: Initialize counts $C[i, j] \leftarrow 0$ and pixel sums $S[i, j] \leftarrow 0$ for all $(i, j)$
4: **for** $t = 1$ to $B$ **do**
5:      Observe frame $o_t$ and query policy $a \leftarrow \pi(o_t)$
6:      $S_t \leftarrow \emptyset$
7:      **for** each cell $(i, j)$ in the $H \times W$ grid **do**
8:          $\tilde{o} \leftarrow \text{INPAINT}(o_t, \text{cell}(i, j), r)$
9:          $\tilde{a} \leftarrow \pi(\tilde{o})$
10:          **if** $\tilde{a} \neq a$ **then**
11:              $S_t \leftarrow S_t \cup \{(i, j)\}$
12:          **end if**
13:      **end for**
14:      **if** $|S_t| = 1$ **then**                          ▷ keep the frame only when exactly one cell flips the action
15:          $(i, j) \leftarrow$ the unique element of $S_t$
16:          $C[i, j] \leftarrow C[i, j] + 1$
17:          $S[i, j] \leftarrow S[i, j] + \text{CROP}(o_t, \text{cell}(i, j))$
18:      **end if**
19: **end for**
20: **if** $\max_{i,j} C[i, j] = 0$ **then**
21:      **return** no location prior
22: **else**
23:      $(i^\star, j^\star) \leftarrow \arg\max_{i,j} C[i, j]$
24:      $\bar{P} \leftarrow S[i^\star, j^\star]/C[i^\star, j^\star]$
25:      **return** $(i^\star, j^\star)$, $C$, $\bar{P}$
26: **end if**

---

### H.2 Algorithmic Details of Mitigation

Recall from Section 4.2 that the mitigation process is summarized in Algorithm 2 in the main text. Here we further provide the more detailed description about the danger state labeling used to expand detection coverage in Algorithm 4. Moreover, a complete version of the replanning procedure for backdoor mitigation and the procedure for generating Trojan rollouts are provided in Algorithm 5 and Algorithm 6, respectively.

---

**Algorithm 4** Danger State Marking in Detection Tree $\mathcal{D}$

---
1: **Input:** Detection Tree $\mathcal{D}$, Leaf node $s_L$, Backtrack depth $K$
2: **Output:** Updated detection tree $\mathcal{T}^{\text{det}}$ with danger states marked
3: $S_{\text{danger}} \leftarrow \{s_L\}; \quad s \leftarrow s_L$
4: **for** $i = 1$ to $K$ **do**
5:      **if** $s$ has parent $s_p$ **then**
6:          Mark $s_p$ as danger
7:          $S_{\text{danger}} \leftarrow S_{\text{danger}} \cup \{s_p\}$
8:          $s \leftarrow s_p$                                   ▷ Move up to parent
9:      **else**
10:          **break**
11:      **end if**
12: **end for**
13: **return** $\mathcal{D}$ (with nodes in $S_{\text{danger}}$ marked)

---

---

**Algorithm 5** MCTS Replan $(s_t, \pi^{\text{Trojan}}, \pi^{\text{adv}}, N, H)$

---

1: **Input:** Current state $s_t$, Trojan policy $\pi^{\text{Trojan}}$, Default rollout adversary policy $\pi^{\text{adv}}$, Simulation budget $N$, Horizon $H$
2: **Output:** Replanned action $a_t^{\text{Trojan}}$
3: Initialize a search tree $\mathcal{T}$ with root node $n_0 \leftarrow s_t$
4: **for** $i \leftarrow 1$ to $N$ **do**
5:     **// Selection and Simulation**
6:     $n \leftarrow n_0$, $h \leftarrow 0$, $path \leftarrow []$
7:     **while** $h < H$ **and** not terminal$(n)$ **do**
8:         **if** $i = 1$ **then**
9:             $a^{\text{Trojan}} \leftarrow \pi^{\text{Trojan}}(n)$
10:         **else**
11:             $a^{\text{Trojan}} \leftarrow \pi^{\text{Trojan}}(n) + \mathcal{N}(0, \sigma^2)$
12:         **end if**
13:         $a^{\text{adv}} \leftarrow \pi^{\text{adv}}(n)$
14:         $(r, n') \leftarrow \text{APPLYACTION}(n, a^{\text{Trojan}}, a^{\text{adv}})$
15:         Append $(n, a^{\text{Trojan}}, r)$ to $path$
16:         $n \leftarrow n'$, $h \leftarrow h + 1$
17:         **if** $h > h_{\text{rollout}}$ **then**
18:             $R_{\text{rollout}} \leftarrow \text{TROJANROLLOUT}(n, \pi^{\text{Trojan}}, \pi^{\text{adv}}, H - h)$
19:             **break**
20:         **end if**
21:     **end while**
22:     **// Backup**
23:     $G \leftarrow 0$
24:     **if** $h \geq h_{\text{rollout}}$ **then**
25:         $G \leftarrow R_{\text{rollout}}$
26:     **end if**
27:     **for** $(\tilde{n}, \tilde{a}, \tilde{r}) \in \text{REVERSE}(path)$ **do**
28:         $G \leftarrow \tilde{r} + \gamma \cdot G$
29:         $\hat{Q}(\tilde{n}, \tilde{a}) \leftarrow \max\big(\hat{Q}(\tilde{n}, \tilde{a}), G\big)$
30:     **end for**
31: **end for**
32: $a_t^{\text{Trojan}} \leftarrow \arg\max_a \hat{Q}(n_0, a)$
33: **return** $a_t^{\text{Trojan}}$

---

---

**Algorithm 6** TrojanRollout$(n, \pi^{\text{Trojan}}, \pi^{\text{adv}}, \Delta)$

---

1: **Input:** Node $n$, Trojan policy $\pi^{\text{Trojan}}$, Default rollout adversary policy $\pi^{\text{adv}}$, remaining horizon $\Delta$
2: **Output:** Estimated return $R$
3: $R \leftarrow 0$, $\gamma' \leftarrow 1$
4: **for** $j \leftarrow 1$ to $\Delta$ **do**
5:     **if** terminal$(n)$ **then**
6:         **break**
7:     **end if**
8:     $a^{\text{Trojan}} \leftarrow \pi^{\text{Trojan}}(n)$
9:     $a^{\text{adv}} \leftarrow \pi^{\text{adv}}(n)$
10:     $(r, n') \leftarrow \text{APPLYACTION}(n, a^{\text{Trojan}}\, a^{\text{adv}})$
11:     $R \leftarrow R + \gamma' \cdot r$
12:     $\gamma' \leftarrow \gamma' \cdot \gamma$
13:     $n \leftarrow n'$
14: **end for**
15: **return** $R$

---

