# OpenReview forum: "Plan2Cleanse: Test-Time Backdoor Defense via Monte-Carlo Planning in Deep Reinforcement Learning"
_TMLR — Accepted by TMLR_

### Review · Reviewer_mwAD · 2025-11-23

**Summary Of Contributions:**

## **Key Strengths**

* Fully black-box and training-free, requiring no access to model weights, gradients, or clean datasets.
* Effectively handles long-horizon, temporally extended triggers better than prior test-time defenses.
* Demonstrates strong and robust empirical performance across MuJoCo, O-RAN, and Atari RL domains.
* Achieves significantly more efficient trigger detection, requiring far fewer environment interactions than PolicyCleanse.
* Provides a simple, lightweight, and practical test-time mitigation module that preserves benign behavior.


## **Key Weaknesses**


- Assumes access to an environment simulator for test-time rollouts, which may be unrealistic in some deployed RL systems.

- Relies on reward degradation as the detection signal, which may be noisy or ambiguous in certain RL tasks.

- Depends strongly on the correctness of the reward design; inaccurate or sparse/delayed rewards can mislead MCTS and cause false positives.

- Uses short-horizon, local replanning (e.g., H=5) that may not guarantee long-term optimal behavior, especially in tasks requiring long-range consistency.

- Requires runtime overriding of the agent’s actions, which may not be feasible in all real-world deployments.

- Assumes adversarial inputs (e.g., opponent actions) are controllable, which may be stronger than some real-world threat models.

**Audience:**

Yes

**Audience Explanation:**

Yes. This paper would attract interest from a meaningful portion of TMLR’s audience because it addresses a growing and underexplored security threat in reinforcement learning (RL): backdoor attacks, especially those involving temporally extended triggers. As shown throughout the paper , RL models are increasingly deployed in high-stakes settings (e.g., control systems, communication networks, robotics), but existing defenses either require access to model weights or fail against long-horizon triggers.

These contributions are directly relevant to ML researchers focused on RL safety, robustness, secure deployment, and model evaluation — all of which are recurring themes in TMLR’s readership. The practical, model-agnostic nature of the defense also aligns with the increasing interest in foundation-model-style RL agents, where white-box assumptions no longer hold. In short: the method fills a clearly recognized gap and will be useful to researchers studying both RL algorithms and ML security.

**Broader Impact Concerns:**

This work primarily proposes a defensive test-time method for detecting and mitigating backdoor attacks in RL. As such, it poses minimal negative societal impact.
The only potential concern is that the trigger-search component could, in principle, provide insights that an adaptive adversary might abuse to design stronger triggers, but this risk is limited and secondary to the paper’s clear defensive focus.
Overall, the work strengthens RL security and does not raise major broader impact concerns.

**Claims And Evidence:**

No

**Claims Explanation:**

My evaluation is based on a detailed examination of the paper’s methodology, assumptions, and experimental evidence. The strengths follow naturally from the design choices articulated in the paper. For instance, the framework is explicitly positioned as a fully test-time, training-free defense that does not require policy retraining, gradient access, or clean datasets, and this is reinforced across the methodology and experiment sections. The paper demonstrates that this black-box approach is not only feasible but also effective across MuJoCo, O-RAN, and Atari, which are structurally different RL domains in terms of state dimensionality, action semantics, and trigger mechanisms. These results provide strong support for the generality and robustness claims made by the authors.

Moreover, the paper’s key idea—to recast backdoor detection as a trajectory-level planning and search problem rather than a learning problem—is both conceptually clear and empirically validated. Pages 6–7 describe how the method evaluates candidate triggers using negated cumulative reward and systematically explores adversarial sequences using MCTS, without needing a probing policy or white-box analysis. This search-based framing allows the method to handle long-horizon, temporally extended triggers, which are difficult for supervised-learning–style defenses or single-step perturbation methods. The experiments consistently show higher trigger detection success rates and lower interaction overhead compared to prior test-time approaches like PolicyCleanse.

However, the paper also has several limitations originating from its design assumptions and operational requirements. The detection stage relies heavily on reward degradation as the signal for identifying potential triggers (page 6). This works well in benchmarks such as Humanoid or O-RAN where reward signals degrade sharply when a backdoor activates, but may fail in environments with sparse or highly stochastic rewards. In such settings, natural fluctuations may resemble malicious degradation, weakening the reliability of the detection score.

The mitigation strategy further assumes the ability to perform localized test-time replanning via MCTS whenever the input falls into the “dangerous” regions mapped during detection (pages 7–8). This approach uses a short-horizon lookahead (e.g., H=5), which is computationally lightweight but may not guarantee long-term optimal behavior in environments where corrective actions require extended planning. The authors illustrate that their method can efficiently neutralize short-term deviations, but the lack of long-range foresight could limit applicability in high-latency or hierarchical tasks where backdoor effects compound more slowly.

Another key assumption is that defenders have access to an environment simulator, both for exploring candidate triggers and for performing replanning rollouts. This assumption is implicit in the detection and mitigation algorithms (pages 6–8), as both rely on simulated forward transitions to compute Q-values for the search tree. While this is practical for robotics simulators or O-RAN testbeds, it may not hold for real-world deployments where simulators are unavailable, inaccurate, or expensive to use.

Furthermore, the mitigation approach requires the defender to override the agent’s actions in real time, which may not be supported in safety-critical or strictly constrained RL deployments (e.g., autonomous vehicles, industrial control systems). The paper demonstrates such intervention in the O-RAN and MuJoCo simulators, but practical deployment contexts may impose restrictions.

Lastly, the threat model assumes that adversarial inputs—such as opponent action sequences in competitive RL—are controllable or repeatable during detection (page 6). Real-world adversaries may behave stochastically or strategically adapt, making it less realistic to expect defenders to reproduce attack sequences as systematically as the method requires.

**Requested Changes:**

• The authors should provide a deeper analysis of the method’s reliance on reward degradation as the detection signal.
Because Plan2Cleanse uses negated reward to identify trigger activation, its detection quality may degrade in environments with sparse, delayed, or noisy rewards. Such reward signals can cause false positives by misclassifying normal suboptimal behavior as malicious, or false negatives when the backdoor impact is hard to distinguish. Adding experiments in noisy-reward settings or discussing alternative signals beyond raw reward (e.g., state features, auxiliary metrics) would significantly strengthen the reliability of the detection stage.

• The paper should further discuss and evaluate the limitation introduced by short-horizon, local replanning used for mitigation.
The proposed mitigation relies on small planning horizons (e.g., H=5), which corrects behavior only locally. This may be insufficient for tasks that require long-term policy consistency because backdoor effects may accumulate across many steps. The authors should analyze whether deeper planning or hierarchical strategies improve robustness, or clarify scenarios where short-horizon replanning might fail to prevent long-term deviations.

• The authors should clarify the practicality of assuming access to an environment simulator for test-time rollouts.
Plan2Cleanse depends on simulating forward trajectories for both trigger search and online mitigation. However, many real-world RL deployments—such as robotics, industrial control, and wireless networks—do not provide accurate simulators or do not allow arbitrary rollouts. The paper should articulate how the method would operate under partial, noisy, or approximate simulators, or discuss alternative strategies when such simulators are unavailable.

• The detection mechanism would benefit from a discussion on controlling false positives, especially in low-variance reward environments.
Because the method considers a sequence malicious whenever it degrades reward, environments with inherently small reward variability may cause benign sequences to appear dangerous. The paper should include analysis or safeguards that reduce the risk of misclassifying natural fluctuations as backdoor activations.

• The threat model should be better justified, particularly the assumption that adversarial inputs (e.g., opponent actions) are controllable during detection.
In many real-world settings, adversaries do not offer deterministic or fully controllable triggers. If opponent actions or environmental perturbations are stochastic or adaptive, the defender may not be able to systematically search for triggers the same way. The authors should explain how Plan2Cleanse generalizes to uncontrollable adversaries or clarify the limits of the current assumption.

• The paper should expand on how long-term stability is ensured during extended deployment under continuous mitigation.
Although the replanning module is lightweight, repeated intervention may accumulate errors or introduce policy drift over long trajectories. Additional experiments or discussion on the impact of repeated replanning on long-term stability would improve the completeness of the evaluation.

---

> ### Author Response · Authors · 2026-03-01
> **Response to Reviewer mwAD (Part 1)**
>
> We thank the reviewer for the thoughtful and constructive feedback. We address each concern point-by-point below and have revised the paper accordingly.
>
> **[Requested Change 1, Weaknesses 2 and 3]**
> > Adding experiments in noisy-reward settings or discussing alternative signals beyond raw reward (e.g., state features, auxiliary metrics) would significantly strengthen the reliability of the detection stage.
>
> > Relies on reward degradation as the detection signal, which may be noisy or ambiguous in certain RL tasks.
>
> > Depends strongly on the correctness of the reward design; inaccurate or sparse/delayed rewards can mislead MCTS and cause false positives.
>
> Thank you for the thoughtful feedback. As suggested by the reviewer, we conducted additional experiments in MuJoCo (Ant and Humanoid) in both the noisy-reward and sparse-reward settings to evaluate the reliability of Plan2Cleanse in detecting backdoor triggers under noisy rewards and sparse rewards.
>
> (1) **Noisy-reward setting**: We inject zero-mean Gaussian noise $\mathcal{N}(0,\sigma^2)$ into the reward at each time step. This introduces the noise standard deviation $\sigma$ as an additional hyperparameter. All other hyperparameters remain identical to those used in the main experiments (please see Table 4 in Appendix B.1 in the updated manuscript). Reported results are averaged over 30 random seeds.
>
> We report the final TDSR of Plan2Cleanse under various levels of noise standard deviation.
> The results are available at: https://ibb.co/qLDTS20m
>
> We observe that the final TDSR is only minimally affected by the reward perturbation. Moreover, we also found that even in the noisy-reward setting with a large noise variance, Plan2Cleanse still achieves 0 false positives (out of 100,000 episodes). These results demonstrate that Plan2Cleanse can reliably detect the backdoor triggers under inaccurate rewards.
>
> (2) **Sparse-reward setting**: The environment reward is set to 0 at every time step except at the end of each episode: the agent receives +1 if it wins and receives −1 if it loses. This setting introduces no additional hyperparameters. All other configurations follow those of the main experiments (Appendix B.1 in the updated manuscript), and the results are averaged over 30 random seeds.
>
> We report the TDSR of Plan2Cleanse across iterations under sparse rewards as follows: https://ibb.co/0R2K2J6Z
>
> Again, we observe that the final TDSR remains close to 1 like in the original dense-reward setting, showing that Plan2Cleanse can reliably detect the backdoor even with only sparse reward signals.
>
> We have also included the above additional results in Appendix G for completeness.

---

> ### Author Response · Authors · 2026-03-01
> **Response to Reviewer mwAD (Part 2)**
>
> **[Requested Change 2, Weakness 4]**
> > The paper should further discuss and evaluate the limitation introduced by short-horizon, local replanning used for mitigation. The proposed mitigation relies on small planning horizons (e.g., H=5), which corrects behavior only locally. This may be insufficient for tasks that require long-term policy consistency because backdoor effects may accumulate across many steps. The authors should analyze whether deeper planning or hierarchical strategies improve robustness, or clarify scenarios where short-horizon replanning might fail to prevent long-term deviations.
>
> > Uses short-horizon, local replanning (e.g., H=5) that may not guarantee long-term optimal behavior, especially in tasks requiring long-range consistency.
>
> Thank you for this insightful suggestion. We agree that, in principle, increasing the replanning horizon can further improve mitigation performance, especially in tasks that require stronger long-term consistency. In Plan2Cleanse, the replanning horizon $H$ is a tunable hyperparameter that enables a trade-off between mitigation latency, computational overhead, and robustness.
>
> Importantly, replanning in our framework is
> - **Selectively triggered** (only when anomaly detection signals potential backdoor activation)
> - Used as a **lightweight corrective mechanism**, rather than as a persistent controller.
>
> Its purpose is to locally neutralize abnormal deviations when a potential backdoor activation is detected. Empirically, we observe that once such deviations are corrected early, the agent naturally returns to benign trajectories without requiring long-horizon intervention.
> Our experiments show that increasing $H$ beyond a small value brings little additional benefit:
> - **Humanoid**: Figure 8(a) (Section 5.7) shows negligible gains when $H \ge 4$. The figure is also available at https://ibb.co/bgYjHNtC
> - **Ant**: In Appendix F.3, Figure 15(a) similarly shows minimal improvement beyond $H \ge 4$. The figure is also available at https://ibb.co/p63HmGfQ
> - **O-RAN mobile-env**: Figure 16 indicates that the data rate remains nearly unchanged when $H \ge 5$. The figure is also available at https://ibb.co/Xf58HtnW
> - **Atari**: Figure 17 shows only slight score improvement for $H \ge 15$. The figure is also available at https://ibb.co/7J36qVS2
>
>
> Across domains, mitigation performance quickly saturates with moderate horizons, suggesting that short-horizon replanning is sufficient for the studied backdoor settings. While deeper or hierarchical planning could be incorporated in principle, our empirical results indicate diminishing returns relative to the increased computational cost.
> To better clarify this design choice, we have added a paragraph explicitly discussing the horizon trade-off and empirical saturation behavior in Sections 4.2, 5.6, and Appendix F.3.

---

> ### Author Response · Authors · 2026-03-01
> **Response to Reviewer mwAD (Part 3)**
>
> **[Requested Change 3, Weakness 1]**
> > The authors should clarify the practicality of assuming access to an environment simulator for test-time rollouts. Plan2Cleanse depends on simulating forward trajectories for both trigger search and online mitigation. However, many real-world RL deployments—such as robotics, industrial control, and wireless networks—do not provide accurate simulators or do not allow arbitrary rollouts. The paper should articulate how the method would operate under partial, noisy, or approximate simulators, or discuss alternative strategies when such simulators are unavailable.
> > Assumes access to an environment simulator for test-time rollouts, which may be unrealistic in some deployed RL systems.
>
> Thank you for raising this important point regarding simulator access and practical deployment.
>
> First, we would like to clarify that our backdoor detection component is designed primarily for a **pre-deployment setting**. In this phase, it is standard practice in robustness and safety evaluation to stress-test learned policies in controlled environments before real-world release (Lee et al., 2020; Corso et al., 2021; Sidrane et al., 2022). The assumption of access to a high-fidelity simulator during detection is therefore consistent with common evaluation pipelines in safety-critical RL applications, where extensive simulation-based validation is routinely performed prior to deployment (Sinha et al., 2020; Wang et al., 2021; Feng, 2023). In this context, simulator access is not an additional burden introduced by our method, but rather aligned with established verification workflows.
>
> Second, for mitigation at test time, we note that the availability of simulators is increasingly realistic due to the widespread adoption of **digital twin** technologies. In many modern applications, including robotics (e.g., Omniverse and Gazebo), industrial control (Xu et al., 2023), and wireless communication systems (Hoydis et al., 2022), it is now common to construct digital replicas of physical systems to enable safe testing, planning, and performance optimization. As a result, access to at least an approximate simulator is often already part of the system infrastructure. Our method does not require a perfect simulator as short-horizon rollouts are used for relative trajectory comparison and local correction, which can tolerate moderate modeling inaccuracies.
>
> We agree that in settings where no simulator or digital twin is available, applying Monte-Carlo planning-based defenses would be more challenging. We clarify this scope in the paper and explicitly discuss that Plan2Cleanse is most suitable for scenarios where pre-deployment validation or digital twin infrastructure is available, which is increasingly common in modern RL deployments.
>
> **References**
> - Lee, Ritchie, et al., "Adaptive stress testing: Finding likely failure events with reinforcement learning," Journal of Artificial Intelligence Research (JAIR), Volume 69, pp. 1165-1201, 2020.
> - Sinha, Aman, et al., "Neural bridge sampling for evaluating safety-critical autonomous systems," Advances in Neural Information Processing Systems (NeurIPS), pp. 6402-6416, 2020
> - Wang, Jingkang, et al., "Advsim: Generating safety-critical scenarios for self-driving vehicles." IEEE/CVF Conference on Computer Vision and Pattern Recognition (CVPR), 2021.
> - Corso, Anthony, et al., "A survey of algorithms for black-box safety validation of cyber-physical systems," Journal of Artificial Intelligence Research (JAIR), Volume 72, pp. 377-428, 2021.
> - Sidrane, Chelsea, et al., "OVERT: An algorithm for safety verification of neural network control policies for nonlinear systems," Journal of Machine Learning Research (JMLR), Volume 23, pp. 1-45, 2022.
> - Feng, Shuo, et al., "Dense reinforcement learning for safety validation of autonomous vehicles." Nature, Volume 615, pp. 620-627, 2023.
> - Hoydis, Jakob, et al., "Sionna: An open-source library for next-generation physical layer research." arXiv preprint arXiv:2203.11854, 2022.
> - Xu, Hansong, et al., "A survey on digital twin for industrial internet of things: Applications, technologies and tools," IEEE Communications Surveys & Tutorials, Volume 25, pp. 2569-2598, 2023.

---

> ### Author Response · Authors · 2026-03-01
> **Response to Reviewer mwAD (Part 4)**
>
> **[Requested Change 4]**
> > The detection mechanism would benefit from a discussion on controlling false positives, especially in low-variance reward environments. Because the method considers a sequence malicious whenever it degrades reward, environments with inherently small reward variability may cause benign sequences to appear dangerous. The paper should include analysis or safeguards that reduce the risk of misclassifying natural fluctuations as backdoor activations.
>
> We thank the reviewer for highlighting the importance of controlling false positives, particularly under low reward variability. We would like to clarify that the core objective of Plan2Cleanse is not merely to flag any reward degradation, but to **leverage Monte Carlo Tree Search (MCTS) to actively explore action sequences that induce systematically undesirable outcomes**. In practice, these outcomes correspond to semantically meaningful failures (e.g., collapse, instability, or task violation), which typically manifest as consistent and substantial reward drops relative to the policy’s expected trajectory.
>
> Moreover, even when the primitive reward values are of small magnitude, the defender retains the flexibility to specify undesirable system behaviors (e.g., instability, task failure, or safety violations), map them to safety- or task-level signals, and incorporate them into an augmented reward. Consequently, detection is not triggered by minor stochastic fluctuations, but by sequences that induce clear and semantically meaningful degradation relative to normal policy behavior. This design ensures that the planning procedure focuses on structurally harmful behaviors rather than reacting to natural reward noise.
>
> Importantly, the use of the negated Trojan reward as the default detection score is not intrinsic to Plan2Cleanse. The framework is agnostic to the specific reward formulation and can operate with any task-dependent metric that captures undesirable or malicious system behavior. This flexibility enables practitioners to incorporate normalization, smoothing, threshold calibration, or domain-specific safety signals when operating in environments with inherently low reward variability. Therefore, **Plan2Cleanse does not fundamentally rely on raw reward magnitude but on the ability to differentiate structured adversarial effects from natural variability**.
>
> Empirically, we do not observe false positives in the evaluated settings. Specifically:
> (1) Across all reported domains (MuJoCo, O-RAN, and Atari), no benign sequences were misclassified as malicious in the original experimental configurations.
> (2) In the sparse-reward setting described in our response to Requested Change 1, both Ant and Humanoid exhibit zero false positive events throughout training.
>
> These results suggest that the planning-based search mechanism does not overreact to natural reward fluctuations, even in environments with limited reward variability. We have included the above discussion in Section 4.1 in the revised manuscript.

---

> ### Author Response · Authors · 2026-03-01
> **Response to Reviewer mwAD (Part 5)**
>
> **[Requested Change 5, Weakness 6]**
> > The threat model should be better justified, particularly the assumption that adversarial inputs (e.g., opponent actions) are controllable during detection. In many real-world settings, adversaries do not offer deterministic or fully controllable triggers. If opponent actions or environmental perturbations are stochastic or adaptive, the defender may not be able to systematically search for triggers the same way. The authors should explain how Plan2Cleanse generalizes to uncontrollable adversaries or clarify the limits of the current assumption.
> > Assumes adversarial inputs (e.g., opponent actions) are controllable, which may be stronger than some real-world threat models.
>
> We would like to clarify that in Plan2Cleanse, **our assumption is aligned with standard test-time backdoor defenses**: the defender has access to a simulator or environment model that allows evaluation of candidate input sequences, but does not control the real adversary during deployment.
>
> **Detection phase**.
> Similar to existing detection approaches, Plan2Cleanse performs detection via an offline, “red-teaming”-like procedure. Specifically, MCTS is used to search for potential backdoor trigger sequences by simulating adversarial interactions within the environment. This search process does not require the real adversary to provide deterministic or controllable triggers at deployment time. Instead, the planner explores the space of possible input or opponent behaviors to identify sequences that could induce systematically undesirable outcomes. In other words, controllability is not assumed for the real-world adversary; it is only the defender’s ability to simulate and evaluate candidate sequences during detection that is required. This is a standard and practical assumption in reinforcement learning settings where environment rollouts are available.
>
> **Mitigation phase**.
> During mitigation, the adversary’s actions are never altered or overridden. As described in Section 4.2, the defense mechanism operates by modifying the Trojan policy’s outputs after a trigger pattern has been detected. The opponent or environment remains unchanged. Therefore, even if adversarial behaviors are stochastic or adaptive at deployment, Plan2Cleanse functions as a reactive safeguard on the policy side.
>
> Overall, Plan2Cleanse does not rely on deterministic or fully controllable adversaries in practice. Rather, it assumes the defender can conduct systematic simulation-based search during detection, while mitigation operates solely on the defended policy. We acknowledge that fully adaptive adversaries, who modify their trigger strategy in response to the defense, represent a more challenging setting that is beyond the current scope, and we discuss this as a limitation and direction for future work.
>
> In the updated manuscript, we have further clarified this distinction in the threat model (Section 3.3) to avoid ambiguity.

---

> ### Author Response · Authors · 2026-03-01
> **Response to Reviewer mwAD (Part 6)**
>
> **[Requested Change 6, Weakness 6]**
> > The paper should expand on how long-term stability is ensured during extended deployment under continuous mitigation. Although the replanning module is lightweight, repeated intervention may accumulate errors or introduce policy drift over long trajectories. Additional experiments or discussion on the impact of repeated replanning on long-term stability would improve the completeness of the evaluation.
>
> We would like to clarify that Plan2Cleanse is designed as a **selective and event-triggered intervention mechanism**, rather than a continuously active controller. As a result, under Plan2Cleanse, long-term stability can be largely achieved for the following reasons:
>
> (1) Replanning is invoked only when a trigger sequence is detected. Under normal operation, the original policy executes without modification. Therefore, mitigation does not repeatedly override the policy along long benign trajectories, which substantially limits the risk of accumulated errors or policy drift.
>
> (2) The replanning module does not update model parameters or permanently alter the learned policy. It operates at inference time and produces localized corrective actions conditioned on the detected trigger. Once the system returns to a normal state, control is handed back to the original policy. As a result, there is no gradient update, memory accumulation, or internal state modification that could compound over time. This design fundamentally differs from continual fine-tuning or adaptive policy modification schemes that may introduce drift.
>
> Notably, in the experiments across all evaluated domains (MuJoCo, O-RAN, and Atari), we do not observe performance degradation along extended trajectories under repeated evaluations. Moreover, clean-task performance remains stable (e.g., Table 3 in Section 5.4 in the updated manuscript), and no progressive instability is observed during long rollouts. These results suggest that the lightweight, event-driven replanning mechanism does not introduce cumulative instability in practice.
>
> We have also added the above discussion on long-term stability in Section 4.2 in the update manuscript. Thank you again for raising this important question.
>
> **[Weakness 5]**
> > Requires runtime overriding of the agent’s actions, which may not be feasible in all real-world deployments.
>
> Thank you for the thoughtful comment. We would like to respectfully justify that the backdoor mitigation approach of Plan2Cleanse is indeed feasible in practice.
>
> (1) Plan2Cleanse does not require unrestricted or permanent overriding of the Trojan policy’s actions. Instead, it performs selective, event-triggered intervention only when a backdoor trigger is detected. Under normal operation, the original policy executes unchanged. Therefore, the defense does not assume continuous external control over the policy, but rather a safeguard mechanism that activates only under suspected malicious conditions. This substantially limits the degree of runtime intervention required in deployment.
>
> (2) Runtime action filtering or safety-layer intervention is a standard mechanism in many real-world deployments, particularly in safety-critical systems such as robotics, networking, and autonomous control (Desai et al., 2019; Chen et al., 2022; Hobbs et al., 2023). In these settings, it is common to include a supervisory module that can validate, adjust, or veto unsafe actions before execution. Plan2Cleanse can be naturally integrated into such an existing supervisory framework, without modifying the underlying policy parameters or the training pipeline. Thus, the required capability “runtime action validation” is already consistent with established system design practices.
>
> (3) Test-time mitigation approaches (such as Plan2Cleanse and PD) involve significantly less intervention than fine-tuning-based mitigation methods (e.g., PolicyCleanse, Neural Cleanse, and BIRD). Importantly, the mitigation module does not retrain, fine-tune, or overwrite the policy parameters. Instead, it provides temporary corrective actions only upon detection of a trigger. As a result, the intervention is localized, reversible, and operationally lightweight compared to methods that permanently alter the model.
>
> **References**
> - Desai, Ankush, et al., "SOTER: A runtime assurance framework for programming safe robotics systems," Annual IEEE/IFIP International Conference on Dependable Systems and Networks (DSN), 2019.
> - Chen, Shengduo, et al., "Runtime safety assurance for learning-enabled control of autonomous driving vehicles," International Conference on Robotics and Automation (ICRA), 2022.
> - Hobbs, Kerianne L., et al., “Runtime assurance for safety-critical systems: An introduction to safety filtering approaches for complex control systems," IEEE Control Systems Magazine, Volume 43, pp. 28-65, 2023.

---

### Review · Reviewer_CYTR · 2025-12-03

**Summary Of Contributions:**

This paper addresses backdoor attacks/defenses in deep reinforcement learning, focusing on both temporally extended triggers in competitive or multi-agent environments and patch-based triggers in image-based Atari tasks. This topic is timely and important to the community. The authors propose Plan2Cleanse, a test-time, black-box defense that does not require access to model parameters, gradients, or training data.

The fundamental idea of the paper is to treat backdoor trigger reveal as a planning problem, which means by using monte carlo tree search  with Voronoi-based exploration strategy, the method actively searches for adversarial action sequences (as it could be for instance suspicious visual regions) that cause high decrease in return. Verified triggers are encoded and represented in a detection tree that marks dangerous state–action regions. At test time, when such regions are encountered, the proposed method performs short-horizon local mcts replanning to select safer actions while keeping the model as is. For Atari, the paper introduces a grid-based occlusion test to localize potential trigger patches using only black-box queries.

Experiments on  MuJoCo environments, a simulated O-RAN wireless network controller, and Atari games show that the method achieves higher trigger-detection success rates, requires fewer environment interactions to discover triggers, and provides competitive or superior mitigation performance compared to prior methods such as PolicyCleanse, Neural Cleanse, Provable Defense (PD), and BIRD (as reported in different plots and tables).

Key strengths:
===========
- A novel formulation of RL backdoor detection as a planning/MCTS problem under a black-box threat model.
- A unified defense that addresses both sequence-based and patch-based triggers.
- Strong empirical evaluation across diverse domains with relevant baselines.
- A practical mitigation strategy that operates at test time without the need to retraining strategy or similar.

Key weaknesses:
==============
- Limited discussion and evaluation of newer backdoor attacks (as for instance post-training or supply-chain attacks) and recent activation-space defenses in RL.
- Some of the trigger verification criteria are very specific to the environment.
- No theoretical guarantees on the coverage or detectability of rare or reward-preserving triggers.

**Audience:**

Yes

**Audience Explanation:**

Yes, because the paper introduces a novel test-time defense for backdoor attacks in reinforcement learning, a topic of clear interest to researchers focused on RL security, adversarial machine learning, and adversarial robustness.

**Broader Impact Concerns:**

The paper main contribution is focusing on defending reinforcement learning systems against backdoor attacks, and it is clearly beneficial for the robustness and safe deployment of RL models. However, several broader-impact aspects could be addressed to make it more explicit. First, the methodology relies on the ability to probe and manipulate the behavior of deployed agents through repeated environment interaction, which may not always be safe in real-world physical systems (e.g., robotics, autonomous driving, industrial control and so on). It is important to discuss how the detection process can avoid causing harm during exploration. Second, while the method aims to improve security, the same technique might help adversaries design more efficient or stealthy backdoors by revealing structural vulnerabilities in reward landscapes or state–action dynamics. Finally, the paper does not address the societal implications of deploying automated test-time action overrides in safety-critical environments, such as who bears responsibility if the replanning mechanism chooses suboptimal or dangerous actions. So, I recommend additng a short discussion to clarify these points and outlining the possible associated safeguards.

**Claims And Evidence:**

Yes

**Claims Explanation:**

The main claims of the paper are supported by clear and convincing evidence in general. The experiments are designed in adequate way, use appropriate metrics, and evaluate the method across three different domains (to mention again MuJoCo, O-RAN, and Atari) , with comparisons to some baselines such as PolicyCleanse, Neural Cleanse, PD, and BIRD. The reported enhancement in trigger detection, efficiency, and mitigation performance are properly documented (for instance, over 0.4 gain in humanoid tdsr and around 0.5 as largest gain in O-RAN case). The main limitation is that the evaluation does not cover some of the most recent backdoor families, such as post-training or supply-chain attacks, which limits, to some extend, the conclusions. Nevertheless, for the threat models actually studied, the evidence provided is coherent, and sufficiently supports the claims made.

**Requested Changes:**

Critical revisions:
=============
1. Broaden and update the related-work discussion on black-box and test-time backdoor defenses:
-----------------------------------------------------------------------------------------------------------------------------------
Several recent and closely related works are not cited or discussed, and including them is important for accurately situating the contribution within the broader backdoor-defense landscape. In particular:

“Mitigating Deep Reinforcement Learning Backdoors in the Neural Activation Space”
Shakti Vyas, Charlie Hicks, Vasilios Mavroudis, IEEE S&P Workshops 2024.

“Beyond Training-time Poisoning: Component-level and Post-training Backdoors in Deep Reinforcement Learning”
S. Vyas, Y. Zhang, R. Jhawar, V. Mavroudis, 2025, arxiv

“REStore: Exploring a Black-Box Defense against DNN Backdoors using Rare Event Simulation”
Quentin Le Roux, Kassem Kallas, Teddy Furon, SaTML 2024.

“Universal Trojan Signatures in Reinforcement Learning”
M Acharya, H Schoelkopf, R Sekharan, NeurIPS 2023 Workshop.

“Fox in the Henhouse: Supply-Chain Backdoor Attacks Against Reinforcement Learning”
Yiming Liu, et al., 2024 ... also this one is still on arxiv

2. Clarify the scope and limitations of the threat model:
-------------------------------------------------------------------------
The current explanation focuses exclusively on training-time backdoors that cause clear reward drops. I think it should be explicitly stated whether the method can address new threat families like post-training backdoors (e.g., TrojanentRL/InfrectroRL) or reward-preserving triggers, and explain why these cases are out of scope or not.

3. Provide a more clear explanation of the detection criteria and the associated generalizability:
------------------------------------------------------------------------------------------------------------------------------
Several used verification rules (for instance, fall detection in Humanoid, MAD thresholding in Ant, etc ...) appear environment-specific. it is better to justify why these rules are representative and discuss if and how they could be adapted to different RL domains.

4. Clarify the computational assumptions during detection:
-----------------------------------------------------------------------------
The proposed method relies on environment rollouts to evaluate suspicious candidate trajectories. The paper should explicitly state the requirement of a high-fidelity simulator, and discuss how this may influence deployment in realistic scenarios where for example, these simulators are not available.


Non-critical revisions:
=================
1. Improve positioning relative to other methods:
-----------------------------------------------------------------
Even if baselines used in experiments are appropriate for the evaluated domains, the paper would benefit from a more explicit comparison (conceptual one) with other RL or general DNN backdoor defenses (for instance, PD, BIRD, SHINE, and activation-space methods), even if direct experimental comparison is not always feasible.

2. Add a short sensitivity analysis of key hyperparameters:
------------------------------------------------------------------------------
The method depends on several parameters (to mention, exploration probability, rollout horizon, Voronoi sampling ...). A brief sensitivity analysis or discussion would help in order to understand robustness and tuning requirements.

3. Discuss the potential risk of missing rare triggers:
---------------------------------------------------------------------
As MCTS explores only a subset of action space, it should be mentioned the possibility of occurenc of undetected low-probability triggers and how this might be mitigated.

---

> ### Author Response · Authors · 2026-03-01
> **Response to Reviewer CYTR (Part 1)**
>
> We thank the reviewer for the thoughtful and constructive feedback. We address each concern point-by-point below and have revised the paper accordingly.
>
> **[Critical Revision 1, Key Weakness 1]**
> > Limited discussion and evaluation of newer backdoor attacks (as for instance post-training or supply-chain attacks) and recent activation-space defenses in RL.
>
> > Broaden and update the related-work discussion on black-box and test-time backdoor defenses: Several recent and closely related works are not cited or discussed, and including them is important for accurately situating the contribution within the broader backdoor-defense landscape. In particular: \
> > [1] “Mitigating Deep Reinforcement Learning Backdoors in the Neural Activation Space” Shakti Vyas, Charlie Hicks, Vasilios Mavroudis, IEEE S&P Workshops 2024. \
> > [2] “Beyond Training-time Poisoning: Component-level and Post-training Backdoors in Deep Reinforcement Learning” S. Vyas, Y. Zhang, R. Jhawar, V. Mavroudis, 2025, arxiv \
> > [3] “REStore: Exploring a Black-Box Defense against DNN Backdoors using Rare Event Simulation” Quentin Le Roux, Kassem Kallas, Teddy Furon, SaTML 2024. \
> > [4] “Universal Trojan Signatures in Reinforcement Learning” M Acharya, H Schoelkopf, R Sekharan, NeurIPS 2023 Workshop. \
> > [5] “Fox in the Henhouse: Supply-Chain Backdoor Attacks Against Reinforcement Learning” Yiming Liu, et al., 2025 ... also this one is still on arxiv
>
> Thank you for these helpful and timely references. We agree that situating Plan2Cleanse more clearly within the evolving landscape of black-box backdoor defenses will strengthen the paper. We have revised the Related Work section accordingly and explicitly discussed [1–5]. Below we also clarify their relationship to our work.
>
> **[1] Mitigating Deep Reinforcement Learning Backdoors in the Neural Activation Space (IEEE S&P Workshops 2024)**
>
> Similar to prior works such as PD and BIRD, [1] studies observation-level backdoor attacks in image-based RL control. In particular, it focuses on in-distribution triggers, i.e., triggers that lie within the anticipated observation distribution and are thus harder to detect. The proposed defense operates in the neural activation space, identifying anomalous neuron activation patterns induced by backdoor triggers. In contrast, Plan2Cleanse does not rely on internal activations or architectural access, but instead performs test-time planning to expose and neutralize malicious behaviors, making it applicable in black-box settings and to temporally extended triggers.
>
> **[2] Beyond Training-time Poisoning: Component-level and Post-training Backdoors in Deep Reinforcement Learning (arXiv 2025)**
>
> This concurrent work introduces two post-training attacks beyond standard training-time poisoning:
> - (i) TrojanentRL, which embeds a backdoor via manipulation of the rollout buffer, and
> - (ii) InfrectroRL, which injects backdoor behavior through direct, data-free modification of pretrained parameters.
>
> Both focus on novel attack methods rather than defenses. Their threat models are complementary to ours. Plan2Cleanse is attack-agnostic and operates at test time, making it potentially applicable to these post-training attacks as well, since it does not assume knowledge of how the backdoor was implanted.
>
> **[3] REStore: Exploring a Black-Box Defense against DNN Backdoors using Rare Event Simulation (SaTML 2024)**
>
> [3] studies black-box backdoor defense in supervised learning and proposes REStore, which leverages rare-event simulation via Monte Carlo sampling to recover triggers. While both REStore and Plan2Cleanse involve Monte Carlo techniques, their objectives and settings differ fundamentally: REStore targets static input-label mappings in supervised DNNs, whereas Plan2Cleanse addresses sequential decision-making in RL and explicitly reasons over temporally extended trajectories during planning. This introduces additional complexity, as the defender must search over combinatorially large sequences of actions rather than static input perturbations, and must account for the dynamic interplay between agent behavior and environment transitions.
>
> **[4] Universal Trojan Signatures in Reinforcement Learning (NeurIPS 2023 Workshop)**
>
> [4] formulates Trojan detection as a meta-classification problem. It trains a meta-classifier to assign a Trojan probability score based on features extracted from observations, assuming access to a labeled dataset of benign and Trojaned agents. In contrast, Plan2Cleanse does **not** require labeled Trojan models, prior trigger knowledge, or retraining. It directly evaluates and mitigates suspicious behaviors online through planning, and can both detect and neutralize backdoor effects.

---

> ### Author Response · Authors · 2026-03-01
> **Response to Reviewer CYTR (Part 2)**
>
> **[5] Fox in the Henhouse: Supply-Chain Backdoor Attacks Against Reinforcement Learning (arXiv 2025)**
>
> [5] proposes a supply-chain backdoor (SCAB) attack in which an adversary with limited access implants backdoors through legitimate interactions with supplied agents or environment components. Like [2], this work expands the attack surface in RL rather than proposing a defense. Plan2Cleanse is orthogonal in focus: it is a test-time defense mechanism that does not assume a specific attack vector and can, in principle, address supply-chain–induced backdoors by identifying anomalous long-horizon behaviors during planning.
>
> **Comparison with Plan2Cleanse in scope:**
>
> - Works such as [1] and [4] focus primarily on detection and rely on activation-space analysis or meta-classification assumptions. Plan2Cleanse instead operates at decision time, requires no labeled Trojan agents, and handles both detection and mitigation.
>
> - [2] and [5] expand the taxonomy of RL backdoor attacks (including post-training and supply-chain settings), while [3] addresses supervised-learning backdoors. These works are complementary in threat modeling, whereas Plan2Cleanse contributes a unified, test-time, planning-based defense mechanism for deep RL.
>
> Importantly, Plan2Cleanse is explicitly designed to address temporally extended triggers and long-horizon behavioral deviations, which are not directly handled in [1], [3], or [4].
>
> **[Critical Revision 2]**
> > Clarify the scope and limitations of the threat model: \
> > The current explanation focuses exclusively on training-time backdoors that cause clear reward drops. I think it should be explicitly stated whether the method can address new threat families like post-training backdoors (e.g., TrojanentRL/InfrectroRL) or reward-preserving triggers, and explain why these cases are out of scope or not.
>
> Thank you for the thoughtful feedback. We agree that clarifying the scope and limitations of our threat model is important. We have revised Section 3.3 to make these points explicit.
> Below we summarize the clarifications.
>
> **Post-training backdoors.**
>
> Plan2Cleanse is agnostic to when the backdoor is implanted. Our detection mechanism operates purely at test time by probing the deployed policy through Monte Carlo planning and identifying action sequences that induce systematically undesirable outcomes. As long as backdoor activation leads to observable behavioral deviation (e.g., unsafe or task-violating trajectories), our method can in principle expose it regardless of whether the backdoor was injected during training or via post-training parameter modification. We now explicitly state that our threat model is insertion-stage agnostic, but behaviorally grounded.
>
> 1. **TrojanentRL**: TrojanentRL poisons the rollout buffer during training. Since Plan2Cleanse does not retrain or fine-tune the target policy, it is unaffected by how the rollout buffer was used during training. Therefore, if the resulting deployed policy exhibits backdoor-triggered behavioral degradation, our method can detect it. The only scenario in which TrojanentRL could bypass our framework is if the defense-time interaction pipeline (e.g., planning rollouts) were itself compromised. Such attacks target the defense infrastructure rather than the learned policy and are outside our scope, which assumes a trusted evaluation environment.
>
> 2. **InfrectroRL**: InfrectroRL implants post-training backdoors by directly perturbing model parameters, with triggers typically realized via pixel-level manipulations in visual observations. This setting is compatible with our observation-level trigger assumption. Our current formulation assumes localized trigger patterns, which is slightly more restrictive. A full empirical evaluation against InfrectroRL-style attacks would require additional experiments, and we identify this as an important and promising direction for future work and clarify this limitation in the revised manuscript.

---

> ### Author Response · Authors · 2026-03-01
> **Response to Reviewer CYTR (Part 3)**
>
> **Reward-preserving triggers.**
>
> We emphasize that **Plan2Cleanse is not designed merely to flag reward drops, but to use MCTS to actively search for trajectories that induce systematically undesirable behaviors**. In our experiments, these behaviors (e.g., instability, collapse, or task violations) naturally correspond to consistent reward degradation relative to the nominal trajectory.
>
> If an attack is strictly reward-preserving under the primitive environment reward, then it is inherently indistinguishable from benign behavior under that metric. In such cases, the defender must define additional safety- or task-level signals (e.g., constraint violations, instability measures) and incorporate them into an augmented reward. Our framework is compatible with this extension: once undesirable behaviors are mapped to measurable penalties, the planning procedure can target them effectively.
>
> Overall, we have revised the manuscript to clearly state that our threat model is behavior-centric, insertion-stage agnostic, and assumes a trusted test-time evaluation pipeline, while acknowledging limitations regarding strictly reward-preserving triggers without auxiliary detection signals.
>
> We now clarify in Sections 3.3 and 4.1 that strictly reward-preserving backdoors under an unmodified reward function fall outside our default assumptions, but can be addressed through domain-specific reward augmentation.
>
> **[Critical Revision 3, Key Weakness 2]**
> > Some of the trigger verification criteria are very specific to the environment.
>
> > Provide a more clear explanation of the detection criteria and the associated generalizability: \
> > Several used verification rules (for instance, fall detection in Humanoid, MAD thresholding in Ant, etc ...) appear environment-specific. it is better to justify why these rules are representative and discuss if and how they could be adapted to different RL domains.
>
> Our experiments span heterogeneous domains (MuJoCo, O-RAN, and Atari), which differ substantially in dynamics, observation spaces, and task semantics. The verification rules used in each environment are instantiations of measurable performance degradation, not intrinsic components of Plan2Cleanse. The core detection mechanism, i.e., **Monte Carlo planning to search for trajectories that induce systematically undesirable outcomes**, is agnostic to the specific degradation metric.
>
> **MuJoCo and Atari**:
> These are widely adopted benchmarks in the RL backdoor literature. To ensure fair and comparable evaluation, we follow established verification protocols used in prior work. In particular, our criteria align with those in PolicyCleanse (Guo et al., ICCV 2023), PD (Bharti et al., NeurIPS 2022), and BIRD (Chen et al., NeurIPS 2023).
>
> Importantly, the rules instantiate two general categories of degradation signals:
> - **Event-based criteria** (e.g., fall detection in Humanoid), which capture catastrophic task failures.
> - **Statistical return-based criteria** (e.g., MAD-based thresholding in Ant), which detect statistically significant deviations from nominal returns.
>
> These two categories represent broadly applicable paradigms (event-triggered failure detection and distributional performance shift detection) rather than environment-specific heuristics.
>
> **O-RAN**:
> In O-RAN, system-level connectivity (e.g., whether a benign UE maintains connection to a BS) is a standard operational metric. We therefore define trigger success in terms of disruption of benign connectivity. This again reflects a domain-relevant, measurable failure condition rather than a method-specific assumption.
>
>
> More generally, Plan2Cleanse requires only that the defender can define a measurable signal indicating undesirable system behavior. This signal may be: \
> (i) environment reward degradation, \
> (ii) violation of safety or task constraints, or \
> (iii) statistically significant deviation from expected performance.
>
> The planning-based detection procedure remains unchanged. Only the evaluation functional differs across domains. We have revised Section 4.1 to clarify that the verification rule is a design choice reflecting domain semantics, not a methodological restriction. In the expanded Appendix C.1, we further articulate how these criteria map to general degradation principles and how they can be adapted to other RL domains by defining appropriate task-level failure or anomaly signals.

---

> ### Author Response · Authors · 2026-03-01
> **Response to Reviewer CYTR (Part 4)**
>
> **[Critical Revision 4]**
> > Clarify the computational assumptions during detection: \
> > The proposed method relies on environment rollouts to evaluate suspicious candidate trajectories. The paper should explicitly state the requirement of a high-fidelity simulator, and discuss how this may influence deployment in realistic scenarios where for example, these simulators are not available.
>
> Thank you for raising this important point. We have revised Sections 1, 3.3 and 4 to explicitly state the requirement of a simulator and justify the feasibility of this setting. Specifically:
> - First, our backdoor detection component is primarily designed for a **pre-deployment evaluation setting**. In safety-critical RL applications, it is standard practice to stress-test learned policies in controlled environments prior to real-world release (e.g., Lee et al., 2020; Corso et al., 2021; Sidrane et al., 2022). Access to a high-fidelity simulator during this phase is therefore consistent with established robustness and safety verification pipelines, where extensive simulation-based validation is routinely conducted (e.g., Sinha et al., 2020; Wang et al., 2021; Feng, 2023). In this context, simulator access is not an additional requirement uniquely imposed by Plan2Cleanse, but rather aligned with common deployment workflows in safety-sensitive domains.
> - Second, the assumption of simulator access is increasingly realistic due to the adoption of **digital twin** technologies. In modern RL-driven systems, such as robotics (e.g., Omniverse, Gazebo), industrial control, and wireless communication networks, it is common to maintain digital replicas of physical systems for safe testing, planning, and performance optimization. In these scenarios, an approximate simulator is already part of the infrastructure. Moreover, Plan2Cleanse does not require a perfect simulator: it relies on short-horizon rollouts for relative trajectory comparison and local behavioral correction, which can tolerate moderate model mismatch.
>
> We acknowledge that in settings where no simulator or digital twin is available, Monte Carlo planning-based defenses become more challenging to deploy. We have clarified this scope in the revised manuscript and explicitly state that Plan2Cleanse is most suitable for scenarios involving pre-deployment validation or digital twin infrastructure, conditions that are increasingly common in contemporary RL applications.
>
> **[Non-critical Revision 1]**
> > Improve positioning relative to other methods: \
> > Even if baselines used in experiments are appropriate for the evaluated domains, the paper would benefit from a more explicit comparison (conceptual one) with other RL or general DNN backdoor defenses (for instance, PD, BIRD, SHINE, and activation-space methods), even if direct experimental comparison is not always feasible.
>
> We have added Table 1 (Section 2) in the updated manuscript to give a more explicit comparison of Plan2Cleanse with all baselines and other backdoor defenses. Thank you for the helpful suggestion.
>
> **[Non-critical Revision 2]**
> > Add a short sensitivity analysis of key hyperparameters: \
> > The method depends on several parameters (to mention, exploration probability, rollout horizon, Voronoi sampling ...). A brief sensitivity analysis or discussion would help in order to understand robustness and tuning requirements.
>
> In Appendix G of the original manuscript, we have included a sensitivity analysis section covering multiple key hyperparameters, including planning horizon $H$, rollout threshold $h_\text{rollout}$, and detection depth $T$.
>
> Additionally, during the rebuttal, we have added a sensitivity analysis of the exploration probability used in MCTS. The figure is also available at https://ibb.co/KpjjZvg9.
> Please refer to Appendix F.2 in the updated manuscript for the details.
>
> These results indicate that our method is robust to moderate variations in hyperparameters.

---

> ### Author Response · Authors · 2026-03-01
> **Response to Reviewer CYTR (Part 5)**
>
> **[Non-critical Revision 3]**
> > Discuss the potential risk of missing rare triggers: \
> > As MCTS explores only a subset of action space, it should be mentioned the possibility of occurrence of undetected low-probability triggers and how this might be mitigated.
>
> We acknowledge that MCTS cannot guarantee exhaustive exploration of the action space. Consequently, low-probability triggers may remain undiscovered. This limitation is inherent to most search-based exploration methods and is not unique to our method. However, unlike naive random search, MCTS mitigates this concern through reward-guided tree expansion, which prioritizes exploration toward action sequences that induce performance degradation, making it substantially more likely to discover low-probability triggers within a fixed interaction budget. This can be alleviated by repeating detection under multiple random seeds. Empirically, in challenging environments such as minimal-responsiveness of O-RAN, our method achieves substantially higher trigger detection success rates than baselines under comparable interaction budgets. We have revised Section 4.1 to explicitly state this limitation and corresponding alleviation strategies.

---

### Review · Reviewer_6pab · 2026-01-20

**Summary Of Contributions:**

This paper proposes Plan2Cleanse, a test-time defense framework against backdoor attacks in reinforcement learning. The main contributions are:

- A reformulation of backdoor detection in RL as a trajectory-level planning problem, using Monte Carlo Tree Search (MCTS).
- A test-time mitigation mechanism that performs replanning to neutralize detected attacks without modifying the model.
- Experimental evaluation across MuJoCo competitive environments, O-RAN wireless network simulators, and Atari games, showing improvements over baselines.

The key strength of this paper is the novelty of the problem they tackle: detection and mitigation of sequence-based triggers in RL. The main weaknesses are (1) the mathematical formalism, which is often inadequate,  (2) related to previous point, the clarity of the paper which could be  improved, (3) the lack of details in the experimentation: how are the triggers defined exactly ? In what situation precisely is the trigger detection algorithm ran ?  (4) the fact the real-world applicability of these techniques which seems low.

**Additional Comments:**

- What is n_0 in Q(n_0, a) in algorithm 1?
- What are precisely the triggers used for O-RAN and MuJoCo experiments? The paper should provide concrete examples.

**Audience:**

Yes

**Audience Explanation:**

The problem of defending against backdoor attacks in RL is important. The black-box, test-time setting is practically relevant for scenarios where models are deployed without access to training data or internal parameters. However:

- The real-world applicability of this approach is  limited: the authors assume that there exists a single sequence of dangerous actions, independent from the current state. This assumption makes it easy to identify the trigger offline. However, it seems reasonable to assume that in real world situation, triggers would only work in specific parts of the state space, in which case one would have to implement an online trigger detection mechanism.
- Implementing an MCTS replanning algorithm seems a nice idea, but this replanning algorithm could be used to improve the current policy, in such a way that the MCTS would be less and less required.

**Broader Impact Concerns:**

None. This is a defense-oriented paper that aims to improve security of RL systems.

**Claims And Evidence:**

No

**Claims Explanation:**

While the experimental results are promising, the paper suffers from significant issues in mathematical formalization and overall clarity.

The paper claims to adopt a "standard RL setting with a discounted MDP M=(S,A,T,R,\gamma)". However, this is not a standard single-agent MDP since an adversary might be present. The setting looks more like a zero-sum game, as evidenced by the use of r^(+) and r^(-)=-r^(+). At least, the reward function should integrate both a_trojan and a_adversary, as it is the case in most MDP extensions for games with adversaries (e.g.  Markov Games or Minimax-Q).
For example, in Equation 2 of Section 3.4, the formulation as written corresponds exactly to the standard MDP training objective which it should not. Using the notation R(s, a_trojan, a_adversary) would be more appropriate and would better distinguish this equation from the standard problem formulation.

In section 3.2, math is very unclear. The authors write "pi^Trojan(s) = pi^fail(s) if triggered and pi^Trojan(s) = pi^win(s) otherwise". A policy cannot map the same state to two different outputs. Is there a partition of the state space into S_triggered and S_normal? Does the policy depend on history (making it non-Markovian)? The transition from pi^fail to pi^win requires the agent to have memory, which the standard MDP framework does not permit. This needs rigorous formalization.

In Section 3.4, equations are informal and unclear. I have difficulties making a rigourous mathematical sense out of equation 1: what are the random variables over which the expectation is taken? Does this depend on the current state? Does t=1 mean this only applies at the beginning of an episode? Also, the definition of r_t^(-) does not account for a^adv.

The Voronoi Optimistic Optimization on Trees algorithm, which is central to the paper, is only informally described in sections 4.1 and 4.2 with details deferred to Appendix B. This is a critical component that readers should not have to fetch from external references (Kim 2020) to understand. The presentation in Appendix B.1 remains unclear: when maintaining "a set of previously sampled adversarial actions {a_t^(1)...a_t^(n)}", does this refer to actions from the root to the current node? This should be presented as a complete algorithm in the main body of the paper. Also, a figure should be devoted to explaining the MCTS tree in detail, with voronoi diagrams in each node (figure 1 contains a too brief description of the tree with a lot of other information).

**Requested Changes:**

- Improve math notation, use sounder mathematical formulations (e.g. R(s,a_trigger,a_adversary)).
- Figure 1 attempts to convey too much information. A dedicated figure showing the intuition of tree search using Voronoi diagrams would be valuable, particularly illustrating how Voronoi regions are associated with each node.
- Provide rigorous definitions for Equations (1) and (2) in section 3. Specify the random variables, clarify dependence on current state, and make the adversary's action appear explicitly.
- Move the Voronoi-based MCTS detection algorithm to the main body with complete pseudocode. The reader should be able to understand the core algorithm without consulting external references.

---

> ### Author Response · Authors · 2026-03-01
> **Response to Reviewer 6pab (Part 1)**
>
> We thank the reviewer for the thoughtful and constructive feedback. We address each concern point-by-point below and have revised the paper accordingly.
>
> **[Requested Change 1, Requested Change 3]**
> > Improve math notation, use sounder mathematical formulations (e.g. R(s,a_trigger,a_adversary)).
>
> > Provide rigorous definitions for Equations (1) and (2) in section 3. Specify the random variables, clarify dependence on current state, and make the adversary's action appear explicitly.
>
> > The paper claims to adopt a "standard RL setting with a discounted MDP M=(S,A,T,R,\gamma)". However, this is not a standard single-agent MDP since an adversary might be present. The setting looks more like a zero-sum game, as evidenced by the use of r^(+) and r^(-)=-r^(+). At least, the reward function should integrate both a_trojan and a_adversary, as it is the case in most MDP extensions for games with adversaries (e.g. Markov Games or Minimax-Q). For example, in Equation 2 of Section 3.4, the formulation as written corresponds exactly to the standard MDP training objective which it should not. Using the notation R(s, a_trojan, a_adversary) would be more appropriate and would better distinguish this equation from the standard problem formulation.
>
> > In section 3.2, math is very unclear. The authors write "pi^Trojan(s) = pi^fail(s) if triggered and pi^Trojan(s) = pi^win(s) otherwise". A policy cannot map the same state to two different outputs. Is there a partition of the state space into S_triggered and S_normal? Does the policy depend on history (making it non-Markovian)? The transition from pi^fail to pi^win requires the agent to have memory, which the standard MDP framework does not permit. This needs rigorous formalization.
>
> > In Section 3.4, equations are informal and unclear. I have difficulties making a rigourous mathematical sense out of equation 1: what are the random variables over which the expectation is taken? Does this depend on the current state? Does t=1 mean this only applies at the beginning of an episode? Also, the definition of r_t^(-) does not account for a^adv.
>
> We thank the reviewer for the detailed feedback. We have revised Sections 3.1–3.4 to address all raised concerns. We first sketch the unified mathematical workflow, then clarify each specific point.
>
> **Unified Mathematical Workflow.** The environment is modeled as a two-agent Markov Game $\mathcal{M} = (\mathcal{S}, \mathcal{A}, \mathcal{T}, \mathcal{R}, \gamma)$, where $\mathcal{A} = \mathcal{A}^\text{Trojan} \times \mathcal{A}^\text{adv}$, $\mathcal{T}: \mathcal{S} \times \mathcal{A}^\text{Trojan} \times \mathcal{A}^\text{adv} \to \Delta(\mathcal{S})$, and $\mathcal{R}: \mathcal{S} \times \mathcal{A}^\text{Trojan} \times \mathcal{A}^\text{adv} \to \mathbb{R}$. At each timestep $t$, the Trojan agent selects $a_t^\text{Trojan} \sim \pi^\text{Trojan}(s_t)$, the adversary selects $a_t^\text{adv}$, and the system transitions via $s_{t+1} \sim \mathcal{T}(\cdot \mid s_t, a_t^\text{Trojan}, a_t^\text{adv})$, yielding reward $\mathcal{R}(s_t, a_t^\text{Trojan}, a_t^\text{adv})$.
>
> - **Detection** seeks an adversarial sequence $a^\text{adv}_{1:T}$ that maximizes cumulative *negated* Trojan reward, revealing the trigger:
>
> $$\max\_{a^\text{adv}\_{1:T}}  \mathbb{E}\_{\pi^\text{Trojan}}\left[\sum\_{t=1}^{T} \gamma^{t-1} r\_t^{(-)}\middle| s\_1\right], \quad r\_t^{(-)} := -\mathcal{R}(s\_t, a\_t^\text{Trojan}, a\_t^\text{adv})$$
>
> - **Mitigation** then finds a cleansed policy that maximizes Trojan reward under the identified trigger $a^\text{adv}_{1:T}$:
>
> $$\pi^\text{cleanse} = \arg\max\_{\pi}  \mathbb{E}\_{\pi} \left[\sum\_{t=0}^{H} \gamma^{t} r\_t^{(+)} \middle| s\_0, a^\text{adv}\_{1:T}\right], \quad r\_t^{(+)} := \mathcal{R}(s\_t, a\_t^\text{Trojan}, a\_t^\text{adv})$$
>
> Note that $r_t^{(+)} = -r_t^{(-)}$ simply reflects a sign convention for the *Trojan agent's own reward* across two sub-problems; the adversary's reward is never modeled, so this is **not** a zero-sum game.

---

> ### Author Response · Authors · 2026-03-01
> **Response to Reviewer 6pab (Part 2)**
>
> **Clarifications on Specific Concerns.**
>
> ***Non-standard MDP (Section 3.1).*** We agree that the original framing as a single-agent MDP was imprecise. We have revised Section 3.1 to explicitly adopt a **two-agent Markov Game** with joint action space $\mathcal{A}^\text{Trojan} \times \mathcal{A}^\text{adv}$ and reward $\mathcal{R}(s, a^\text{Trojan}, a^\text{adv})$ as shown above.
>
> ***Piecewise Trojan policy (Section 3.2).*** The Trojan policy is well-defined and Markovian because the state space $\mathcal{S}$ is partitioned into two *disjoint* subsets $\mathcal{S}^\text{normal}$ and $\mathcal{S}^\text{triggered}$, encoding trigger activation status directly into the state. The piecewise definition
>
> $$\pi^\text{Trojan}(s) = \begin{cases} \pi^\text{win}(s), & s \in \mathcal{S}^\text{normal} \\\ \pi^\text{fail}(s), & s \in \mathcal{S}^\text{triggered} \end{cases}$$
>
> maps each state to a unique output leaving no ambiguity. No history or memory is required; the trigger condition is a function of the current state $s$ alone. We have added this clarification explicitly in the revised Section 3.2.
>
> ***Random variables and state dependence (Section 3.4).*** The expectation in the detection objective is taken over state transitions induced by $\mathcal{T}$, the fixed $\pi^\text{Trojan}$, and the candidate $a^\text{adv}_{1:T}$. The starting state $s_1$ is conditioned upon, not averaged over. The index $t=1$ marks the start of the search trajectory and does not restrict detection to episode beginnings, in which any intermediate state may serve as a new root by the Markov property. We have made all random variables and conditioning explicit in the revised equations.
>
> **[Requested Change 2]**
> > Figure 1 attempts to convey too much information. A dedicated figure showing the intuition of tree search using Voronoi diagrams would be valuable, particularly illustrating how Voronoi regions are associated with each node.
>
> In the revision, we have added a new dedicated figure (Figure 2 in the updated manuscript) that illustrates the intuition of our MCTS procedure using Voronoi diagrams. The new figure is also available at https://ibb.co/q3YBzL6s.
>
> Specifically, the figure separates the planning intuition from the full system pipeline and focuses on how continuous action sampling is organized during tree expansion.
>
> Our detection framework builds upon conventional MCTS but extends it to continuous action spaces via Voronoi-based progressive partitioning. Each expanded action node implicitly defines a Voronoi region in the action space, consisting of all points that are closer (under the chosen metric) to that node’s sampled action than to others.
>
> The new figure explicitly visualizes (i) the partition of the continuous action space into Voronoi regions, (ii) the correspondence between tree nodes and their associated regions, and (iii) how progressive refinement concentrates search around promising trajectories. We expect this dedicated illustration significantly improves clarity and makes the geometric intuition behind our tree search substantially more accessible.
>
> Besides, we have also revised the caption of Figure 1 to clearly express the intuition of the whole framework.
>
> **[Requested Change 4]**
> > Move the Voronoi-based MCTS detection algorithm to the main body with complete pseudocode. The reader should be able to understand the core algorithm without consulting external references.
>
> > The Voronoi Optimistic Optimization on Trees algorithm, which is central to the paper, is only informally described in sections 4.1 and 4.2 with details deferred to Appendix B. This is a critical component that readers should not have to fetch from external references (Kim 2020) to understand. The presentation in Appendix B.1 remains unclear: when maintaining "a set of previously sampled adversarial actions {a_t^(1)...a_t^(n)}", does this refer to actions from the root to the current node? This should be presented as a complete algorithm in the main body of the paper. Also, a figure should be devoted to explaining the MCTS tree in detail, with voronoi diagrams in each node (figure 1 contains a too brief description of the tree with a lot of other information).
>
> We have revised the presentation of the Voronoi-based MCTS detection algorithm and moved it to the main body. Thank you for the thoughtful suggestion.

---

> ### Author Response · Authors · 2026-03-01
> **Response to Reviewer 6pab (Part 3)**
>
> **[Concern 1]**
> > The real-world applicability of this approach is limited: the authors assume that there exists a single sequence of dangerous actions, independent from the current state. This assumption makes it easy to identify the trigger offline. However, it seems reasonable to assume that in real world situations, triggers would only work in specific parts of the state space, in which case one would have to implement an online trigger detection mechanism.
>
> We would like to clarify that our approach does **not** assume state-independent triggers. The detection objective is defined *conditionally on the current state* $s\_1$​, and the planner searches for adversarial action sequences that induce return degradation from that specific state. Therefore, triggers are captured whenever their activation causes systematic behavioral deviation, even if they are only effective within localized regions of the state space. The formulation in Sections 3.2 and 3.4 explicitly accounts for state dependency.
>
> We agree that triggers confined to specific subregions of the state space present a more challenging setting. Importantly, Plan2Cleanse is inherently state-conditional and can be executed online from the agent’s current state, enabling localized trigger probing when needed. While exhaustive online search may be computationally intensive, the framework naturally supports adaptive budget allocation (e.g., increasing search depth only when behavioral anomalies are suspected, or leveraging warm-started trees across adjacent states). In addition, our lightweight mitigation module operates without full re-planning and can intervene once suspicious trajectories are identified.
>
> Accordingly, we view Plan2Cleanse as supporting two practical deployment modes.
> - In the **pre-deployment auditing mode**, a larger search budget can be used to systematically probe different regions of the state space and stress-test the policy before release.
> - In the **online safeguard mode**, the method can be applied selectively from the current state with a limited planning budget, triggering replanning-based mitigation only when suspicious behavior is detected. This allows the defender to balance computational cost and protection strength depending on operational constraints.
>
> We have clarified this two-mode usage in Section 4.2 in the revised manuscript to more transparently describe its flexibility and applicability to state-dependent trigger scenarios.
>
> **[Concern 2]**
> > Implementing an MCTS replanning algorithm seems a nice idea, but this replanning algorithm could be used to improve the current policy, in such a way that the MCTS would be less and less required.
>
> We appreciate this insightful observation. Indeed, the replanning returns accumulated during mitigation could, in principle, serve as a training signal to iteratively refine the Trojan policy, essentially framing our MCTS rollouts as a form of online policy improvement. This is a compelling direction that connects to planning-based policy optimization methods such as AlphaZero (Silver et al., 2017).
>
> However, our framework is intentionally designed for the black-box setting, where the defender has no access to the Trojan policy's internal parameters. Translating replanning returns into policy updates would require gradient computation or iterative fine-tuning, both of which fall outside this assumption. We view this as an interesting avenue for future work in settings where white-box access is available, and have added a brief note to the conclusion accordingly.
>
> **Reference**
>
> Silver et al., "Mastering Chess and Shogi by Self-Play with a General Reinforcement Learning Algorithm," 2017
>
> **[Additional Comment 1]**
> > What is n_0 in Q(n_0, a) in algorithm 1?
>
> $n_0$ denotes the root node of the MCTS tree and corresponds to the current state $s_t$ where the replanning is initiated.
>
> **[Additional Comment 2]**
> > What are precisely the triggers used for O-RAN and MuJoCo experiments? The paper should provide concrete examples.
>
> In the competitive MuJoCo environments, triggers include manually designed motion patterns (e.g., bending two arms in Humanoid) and randomly sampled action subsequences. In O-RAN, triggers are implemented as injected physical-layer signal patterns. For Atari (Pong and Breakout), we follow the SleeperNets protocol to implant patch-style triggers during training. The details of the triggers are also provided in Appendix B.1 in the updated manuscript.

---

### Comment · Action_Editor_xLdS · 2026-02-19
**Response Deleted**

Dear Authors,

It appears that you may have submitted a set of responses to the reviewers which are no longer visible. Could you please confirm whether this was intentional, or if it might be a technical issue with OpenReview? If it was unintentional, we would appreciate it if you could kindly repost your responses before we ask the reviewers to provide a final assessment of the paper.

Thank you in advance.
Best regards,
The AE

---

> ### Author Response · Authors · 2026-02-20
>
> Dear AE,
>
> Thank you for reaching out and for bringing this matter to our attention.
>
> We would like to clarify that an earlier version of our draft responses was inadvertently posted on OpenReview before it was finalized. As the responses were incomplete, we removed them promptly to avoid potential confusion.
>
> We are currently consolidating and refining our feedback to ensure a comprehensive and carefully prepared reply to the reviewers’ comments. We expect to post the complete response within the coming week.
>
> We sincerely apologize for any confusion this may have caused and greatly appreciate your patience and understanding.
>
> Best regards,
>
> The Authors

---

### Comment · Reviewer_CYTR · 2026-02-20
**Official comments deleted**

Official comments deleted several times

---

### Decision · Action_Editor_xLdS · 2026-04-13

**Recommendation:** Accept as is

**Audience:**

Yes

**Audience Explanation:**

A key strength of this work lies in the originality of the problem it addresses as well as its clear empirical improvements over baseline methods. The TMLR audience interested in backdoor attacks will find this paper both relevant and engaging.

**Claims And Evidence:**

Yes

**Claims Explanation:**

Upon submission, the reviewers expressed some concerns regarding the mathematical soundness of the paper. Nevertheless, the authors have done a commendable job addressing all comments and questions, resulting in a clear improvement in the overall quality of the manuscript.